# Soil fungi remain active and invest in storage compounds during drought independent of future climate conditions

Alberto Canarini [1,2,9] ✉, Lucia Fuchslueger [1,3,9] ✉, Jörg Schnecker [1], Dennis Metze [1,4], Daniel B. Nelson [5], Ansgar Kahmen[5], Margarete Watzka[1], Erich M. Pötsch[6], Andreas Schaumberger [6], Michael Bahn [7] & Andreas Richter [1,8] ✉

Microbial growth is central to soil carbon cycling. However, how microbial communities grow under climate change is still largely unexplored. Here we use a unique field experiment simulating future climate conditions (increased atmospheric $CO_2$ and temperature) and drought concomitantly and investigate impacts on soil microbial activity. We trace $^2H$ or $^{18}O$ applied via water-vapor exchange into membrane (and storage) fatty acids or DNA, respectively, to assess community- and group-level adjustments in soil microbial physiology (replication, storage product synthesis, and carbon use efficiency). We show that, while bacterial growth decreases by half during drought, fungal growth remains stable, demonstrating a remarkable resistance against soil moisture changes. In addition, fungal investment into storage triglycerides increases more than five-fold under drought. Community-level carbon use efficiency (the balance between anabolism and catabolism) is unaffected by drought but decreases in future climate conditions, favoring catabolism. Our results highlight that accounting for different microbial growth strategies can foster our understanding of soil microbial contributions to carbon cycling and feedback on the climate system.

Plant inputs and soil organic matter decomposition have been considered as the main drivers controlling the soil organic carbon (SOC) balance, but more recently, growth and biomass production rates of soil microbial communities have been identified as main contributors to SOC formation and persistence[1–3]. In particular, the efficiency with which microbes allocate carbon (C) to growth relative to total C uptake (i.e., microbial carbon use efficiency or CUE) has been suggested to be a key parameter for predicting SOC stock and SOC feedback in response to a changing climate[4]. Improving our ability to accurately quantify soil microbial growth will help to better predict microbial CUE, and is essential to understand the controls and to decipher the mechanisms behind terrestrial C and nutrient cycling[5,6].

Because soil microbial communities respond sensitively to climate change, shifts in microbial physiology can cause large repercussions on global C and nutrient cycles[6–8]. Elevated atmospheric $CO_2$ concentrations ($eCO_2$) induce mostly indirect effects on soil microbes

[1]Centre for Microbiology and Environmental Systems Science, University of Vienna, Vienna, Austria. [2]Department of Biological, Geological and Environmental Sciences, University of Bologna, Bologna, Italy. [3]Environment and Climate Hub, University of Vienna, Vienna, Austria. [4]Doctoral School in Microbiology and Environmental Science, University of Vienna, Vienna, Austria. [5]Department of Environmental Sciences – Botany, University of Basel, Basel, Switzerland. [6]Agricultural Research and Education Centre Raumberg-Gumpenstein, Irdning, Austria. [7]Department of Ecology, Universität Innsbruck, Innsbruck, Austria. [8]International Institute for Applied Systems Analysis (IIASA), Laxenburg, Austria. [9]These authors contributed equally: Alberto Canarini, Lucia Fuchslueger. ✉e-mail: alberto.canarini@hotmail.it; lucia.fuchslueger@univie.ac.at; andreas.richter@univie.ac.at

by stimulating primary productivity[9] and changing C resources available for heterotrophic soil microbes, leading to enhanced microbial growth and decomposition rates[10–12]. Warming can increase plant productivity, but also directly stimulate microbial physiological activity (growth and respiration), decrease microbial CUE and accelerate soil C losses[6,7,13]. Drought, in contrast, reduces accessibility to C and nutrients, and consequently can decrease soil microbial activity[14,15] to the point, where in the absence of water microbes become dormant or die. On an ecosystem level, drought can lead to soil C losses[16], as soil respiration is generally less sensitive to dry conditions compared to plant primary productivity[17,18]. Furthermore, the impact of drought on soil respiration and primary productivity can be modified by eCO$_2$ and warming[19–21]. Assessing soil microbial growth responses to drought and possible interactions with other global change factors in field experiments is therefore crucial. Yet, multifactorial global change experiments conducted under field conditions measuring soil biogeochemical cycling and microbial communities are rare[22], rendering interactive effects on soil microbial feedbacks largely unexplored.

It is notoriously challenging to quantify soil microbial growth, and even more so under drought conditions, without the application of substrate or water[23]. Over the last decade, advances in substrate-free [18]O-based stable isotope probing ([18]O-SIP), tracing [18]O assimilation into newly produced DNA, have improved quantitative measures of physiological rates of microbes on a community-level[24], as well as of taxon-specific bacterial growth rates[25,26]. Recently it was shown that it is possible to enrich soil water with [18]O indirectly via water vapor equilibration, avoiding direct water additions to the soil[27]. This now allows the quantification of soil microbial community-level and taxon-level growth rates of bacteria during drought conditions[27,28]. The link between growth rates and microbial identity is a major focus in microbial ecology and can provide mechanistic insights into the role of soil microbial taxa and community dynamics for biogeochemical C and nutrient cycling. Quantifying taxon-specific growth rates via a DNA (or RNA) based method ([18]O-SIP) has been mainly applied to link bacterial community structure to activity rates. However, it requires a separate analysis to measure fungal activity, which has been largely overlooked. At the same time [18]O-SIP yields a high number of fractioned DNA samples to process and analyze[25,29] which makes it challenging to apply to large sample numbers. Moreover, microbial biomass can increase not only as a result of cell replication, but also through the synthesis of storage compounds[30].

Deuterium ([2]H) incorporation into fatty acids can serve as an alternative SIP approach to measure microbial growth, as all microorganisms synthesize lipids and fatty acids to build and maintain cell membranes regardless of their metabolic strategy and cell cycle stage. [2]H-labelling has been used in single cell studies[31], pure cultures[32] and microbial communities in different environments[33–36]. Lipids likely require less investment into repair mechanisms compared to nucleic acids or proteins, which can be resynthesized during repair and cellular maintenance potentially affecting growth rate estimates[33,37], and provide a more precise measure of microbial membrane production rates and cell growth. The major advantages of tracing [2]H into fatty acids ([2]H-FAME-SIP) are the simultaneous and sensitive quantification of bacterial and fungal replication rates[38]. Although compared to DNA/RNA-based methods, PLFA analysis has a lower taxonomic resolution, it provides a more robust and quantitative estimate of microbial biomass[38].

Another feature of [2]H labelling is that [2]H can be traced into microbial neutral-lipid fatty acids (NLFAs), consisting mainly of triglycerides, which are separated from PLFA after extraction from soil[39]. NLFAs are considered as storage compounds found mainly in fungi and many prokaryotic species, and can account for a large C fraction of the total soil microbial biomass pool[30]. As the NLFAs maintain the same taxonomic specificity as PLFA, they can be used in the same way to differentiate microbial groups[39]. While classically biomass growth is

investigated by measuring cell replication, soil microbes can produce and accumulate large amount of storage compounds representing up to 46% of the total microbial biomass, especially in 'stress' situations[30]. Storage compounds can be particularly relevant in ecosystems experiencing large variations in C and nutrient supply[40,41], such as for instance during drought conditions. The ability of microbial populations to invest resources in storage might enable them to detach their metabolic activity from the immediate resource availability, thereby facilitating a wider range of responses to environmental fluctuations, as previously suggested in a model simulation[40]. During drought conditions dissolved organic C has been shown to accumulate in soil pores, increasing its concentration in the remaining water[15]. At the same time decreased moisture reduces the connection and mobility of substrate to microbes, which might lead to higher C investment into storage compounds. Given the central roles of microbes in soil C cycling, methods that can quantify C invested in microbial growth (i.e., not only cell division, but also storage compound synthesis) are strongly needed, particularly for understanding the consequences of changing climate conditions.

Here, we investigated the responses of soil microbial community growth exposed to multiple global change factors under field conditions. The overarching goal of this work was to quantify changes in community-level and group-specific growth rates of soil microbial communities under drought and potential interactions with simulated future climate conditions, i.e., the combined increase in temperature and atmospheric CO$_2$ concentrations. We hypothesized that: (i) drought reduces microbial community-level growth rates, but the impacts of drought are reduced in a future climate, as both eCO$_2$ and warming may stimulate microbial growth; (ii) fungal and bacterial growth display a different sensitivity to drought, with fungal growth being more resistant compared to bacterial growth; and (iii) soil microbes increase the partitioning of C towards reserve compounds during drought. In a multifactorial global change experiment, we jointly manipulated atmospheric CO$_2$ concentrations (+300 ppm above ambient) and temperature (+3 °C above ambient; termed 'future climate condition' hereafter) and investigated how this condition altered the effects of drought compared to ambient controls (ambient, drought, future climate, future climate + drought). We applied [2]H-FAME-SIP by water vapor equilibration[27] to measure soil microbial growth rates under drought conditions and compared them to [18]O-DNA-SIP obtained with the same technique[27]. These methods are referred to as [2]H-vapor-FAME-SIP and [18]O-vapor-DNA-SIP, respectively, hereafter. This allowed us to link the coarse taxonomic resolution of microbial groups with a quantitative measure of growth rates and storage compound formation, and to compare PLFA-based to DNA-based growth rate estimates. We show that (i) soil microbial communities can maintain half of their growth rates under drought and quickly return to control levels after drought, indicating strong resilience; (ii) soil microbial CUE is insensitive to drought, but decreases under future climate conditions; (iii) fungi display remarkable resistance to drought, with no effect of drought on growth rates, and an increase in the investment into storage compounds, accounting for about three-fold the investment into growth. We further discuss the implications of microbial community responses to drought for soil biogeochemistry under future global change scenarios.

## Results

### Soil microbial communities maintain half of their growth rates during drought but recover quickly

The drought treatment significantly reduced soil water content throughout the experimental period (Supplementary Fig. 1). At the time of the peak drought sampling, gravimetric soil water content was 32.0% and 31.0% in ambient and future climate conditions compared to 8.0% and 6.2% in the respective drought treated plots (Supplementary Table 1). Under both ambient and future climate drought significantly

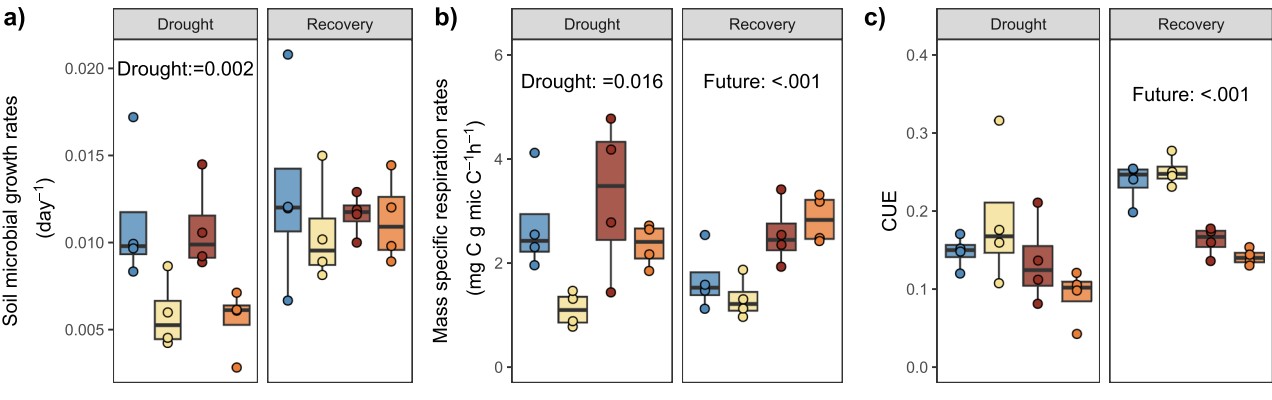

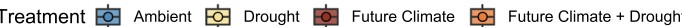

**Fig. 1 | Soil microbial community-level growth rates, mass specific respiration and carbon use efficiency (CUE) during peak drought and recovery. a** Soil microbial growth rates obtained via $^2$H incorporation into PLFA, **b** mass specific respiration rates, and **c** microbial CUE, measured at peak drought ('Drought') and after field rewetting ('Recovery'). Significant differences ($p < 0.05$) between treatments derived from linear mixed models are reported in the figure (the full report is provided in Supplementary Table 2). Box centre line represents median, the box indicates the upper and lower quartiles, whiskers the 1.5x interquartile range, and separated points represent potential outliers. The sample size '$n$' represents biologically independent samples ($n = 4$ replicates). Colour indicates treatment. Source data are provided with this paper.

reduced PLFA-based soil microbial growth rates (Fig. 1a) and mass specific growth rates by 48% ($p = 0.001$, Supplementary Table 2, Supplementary Fig. 2f). Similarly, respiration rates significantly decreased in response to drought (Fig. 1b and Supplementary Fig. 2). Drought did not significantly change microbial community level CUE; however, the future climate treatment caused a significant reduction in CUE during the recovery period only ($p = 0.001$; Supplementary Table 2, Fig. 1c and Supplementary Fig. 2). Microbial community turnover time obtained by the PLFA approach changed from 96 days (rounded values) in both ambient and future climate, to 184 and 204 days during drought, respectively. During the recovery period, microbial community turnover times were similar among all treatments (ambient: 91 days, drought: 100 days, future climate: 87 days, future climate + drought: 92 days).

Mass specific growth rates and CUE calculated via $^2$H incorporation into PLFAs were consistently lower (-50%) compared to the mass specific growth obtained by $^{18}$O incorporation into DNA, but growth rates significantly correlated and the relative differences induced by the climate change treatments remained constant (Supplementary Table 2, Supplementary Fig. 2). Similarly, microbial community turnover time calculated via the $^{18}$O method was lower than with the PLFA approach. At peak drought values changed from 39 days (rounded values) in both ambient and future climate, to 91 and 118 days during drought, for ambient and future climate respectively. In the recovery period, turnover rates in ambient conditions were 45 and 47 days, in non-drought and droughted plots, respectively, while in future climate conditions rates were 35 and 34 days, in non-drought and droughted plots, respectively.

### The active microbial community changes in response to drought and future climate

Our $^2$H-vapor-FAME-SIP approach to trace $^2$H into different PLFA biomarkers allowed not only tracking microbial activity (i.e., growth and CUE) at the community level, but also distinguishing the production of different fatty acids, indicative of the activity of different microbial groups during drought. Drought, future climate and their interaction changed the active microbial community (displayed as principal component analysis, Fig. 2, permanova results in Supplementary Table 3) at peak drought. Drought separated the active community on

PC1 (43%), dominated by a high incorporation of $^2$H into fungal markers, relative to a low incorporation into gram-positive markers (Fig. 2). Future climate conditions favored the incorporation of $^2$H into gram-negative and actinobacterial makers on PC2 (30.3%). Total PLFA abundance (not $^2$H incorporation) showed similar, but weaker patterns (Supplementary Table 3 and Supplementary Fig. 3). The impact of the preceding drought was maintained in the recovery period and significantly changed the relative $^2$H incorporation into PLFA biomarkers (Fig. 2 and Supplementary Table 3) and relative PLFA abundances (Supplementary Fig. 3 and Supplementary Table 3). In the recovery period, future climate only significantly affected the relative abundance, but not the $^2$H incorporation into PLFAs (Fig. 2, Supplementary Fig. 2 and Supplementary Table 3). Comparison between absolute PLFA abundances and mass-specific growth rates shows that higher abundance did not correspond to higher mass-specific growth rates (Supplementary Fig. 4).

### Fungal mass specific growth rates display remarkable drought resistance

Mass specific growth rates of gram positive, gram negative and actinobacteria markers (as well as general microbial markers) significantly decreased with drought, while fungal rates were not affected (Fig. 3, Supplementary Fig. 5 and Supplementary Table 4). At peak drought, gram-positive markers decreased on average by 49% in soils from ambient and 53% in future climate treatments, while gram negative markers decreased by 54% and 49%, respectively. In contrast, actinobacteria biomarkers decreased less (22% in soils from ambient and 44% in future climate treatments; Supplementary Fig. 5). The mass specific growth rate of fungi relative to bacteria was significantly increased under drought conditions by 84% and 173% in ambient and future climate, respectively (Fig. 3 and Supplementary Table 4). The fungi to bacteria mass-specific growth returned to control values after rewetting (Fig. 3d). During the recovery period mass specific growth rates of all microbial groups became similar to levels measured in ambient soils (Fig. 3), with no significant effect of treatments.

Mass specific growth rates correlated positively with respiration rates (Supplementary Fig. 6). Actinobacteria and gram-negative bacteria displayed the strongest degree of correlation ($r = 0.87$ and $r = 0.85$, respectively), while fungi had the lowest degree of correlation

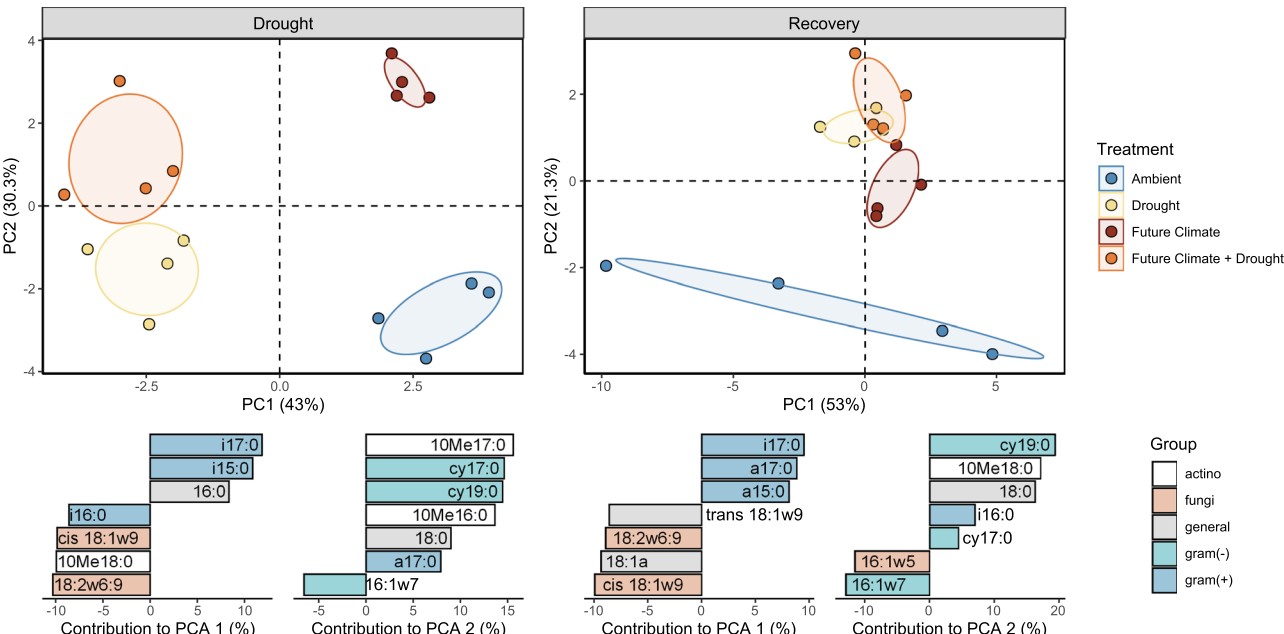

**Fig. 2 | Multivariate analysis of ²H incorporation into individual PLFA biomarkers.** PCA analysis of ²H incorporation into individual PLFA biomarkers (mass specific rates) during peak drought ('Drought') (left panel) and after field rewetting ('Recovery'). Bottom graphs represent the relative contribution (in %) of the top seven variables to the principal components (the positive or negative sign indicates the direction along the respective PCA axes) assigned to microbial groups (gram positive: blue; gram negative: light blue; fungi: orange; actinobacteria: white;

general markers: grey; the arbuscular mycorrhiza fungal biomarker 16:1ω5 is included in the fungi group in the multivariate analysis but not in other graphs displaying fungi. Permanova results are reported in Supplementary Table 3. The sample size represents biologically independent samples ($n$ = 4 replicates). Ellipses represent the 95% confidence intervals. Colour indicates treatment. Source data are provided with this paper.

($r$ = 0.43) and the fungi to bacteria ratio correlated negatively with respiration rates ($r$ = −0.38). As PLFAs are assumed to be a good indicator of viable biomass, we also compared results obtained with our SIP approach to results obtained by using absolute abundance of PLFA values. We found different results, with no overall significant effects except for future climate in gram positive and a significant effect of drought and rewetting on the fungi to bacteria ratio (Supplementary Fig. 7). Absolute growth rate (ng C g⁻¹ soil h⁻¹) pattern were almost identical results as shown from the mass-specific values (Supplementary Fig. 8). Correlation of abundance values with respiration rates were all positive, but displayed lower correlation coefficients, compared to correlations between mass specific growth rates and respiration rates (Supplementary Fig. 9), and fungal abundance displayed the highest correlation coefficient with respiration rates.

### Fungi increase the synthesis of storage compounds during drought

NLFAs represent reserve compounds, which were detectable in fungi-associated fatty acid markers (18:1ω9cis and 18:2ω6,9), the marker 16:1ω7 (a biomarker for gram-negative bacteria), 16:1ω5 (arbuscular mycorrhizal fungi) and the general markers 16:0, 18:0 and 18:1ω9trans. We found that during drought the new production of fungal NLFAs was significantly higher compared to non-drought treated soils (Fig. 4a). The ratio of fungal NLFA to PLFA biomarkers (expressed as percentage) was on average 228 and 305% during drought, but only 22% under ambient and non-drought treated future climate (Fig. 4b). NLFA production rates decreased to ambient levels in the 'Recovery' period. Similar patterns were found for gram-negative bacterial markers, but with a smaller increase of the respective NLFA relative to PLFA marker production during drought (up to 6.3%; Supplementary Fig. 10 and Supplementary Table 5). The total amount of fungal NLFA also increased with drought (Supplementary Fig. 10), but with a stronger increase in future climate conditions (142% in ambient conditions vs 268% in future climate conditions; Supplementary Table 6). Total NLFA

production also increased with drought but only in future climate conditions and relatively less than fungal NLFA (Supplementary Fig. 10 and Supplementary Table 6).

## Discussion

Soil microbial biomass growth ultimately determines soil C and nutrient cycling and is thus a key variable in microbial ecology[1–3]. Despite the significance of microbial growth dynamics for predicting the impacts of global environmental changes on soil biogeochemistry[1,6–8], there is still little direct evidence of responses of microbial growth rates under field conditions during drought and predicted future climate scenarios. Here, we tackled this knowledge gap by using ²H-vapor-FAME-SIP that allows the concomitant estimation of community-level and group-specific C allocation to biomass growth and to storage product synthesis, as well as community-level C use efficiency. We applied ²H-vapor-FAME-SIP to responses of soil microbial communities to simulated drought and future climate conditions (eCO₂ and warming) within a field experiment in a montane grassland. We show that drought strongly reduces bacterial growth, whereas fungi display a remarkable resistance to drought even in a future climate scenario.

Drought decreases the water filled pore space of soil, which constrains connectivity and C and nutrient resource diffusion, whereas in the remaining water filled pore space osmotic potential strongly decreases[14]. In our experiment the simulated drought significantly reduced soil water content (by 73% on average) and, in accordance with our hypothesis, community-level mass-specific growth rates decreased by half. This is in line with reports from previous laboratory incubations observing reduced microbial growth with lowering soil moisture[42,43]. However, in many previous studies, substrates were applied dissolved in water, and could have triggered short-term rewetting responses causing an over-estimation of microbial growth activity under very dry conditions[42–48]. On the other hand, our results show that 50% of the microbial community

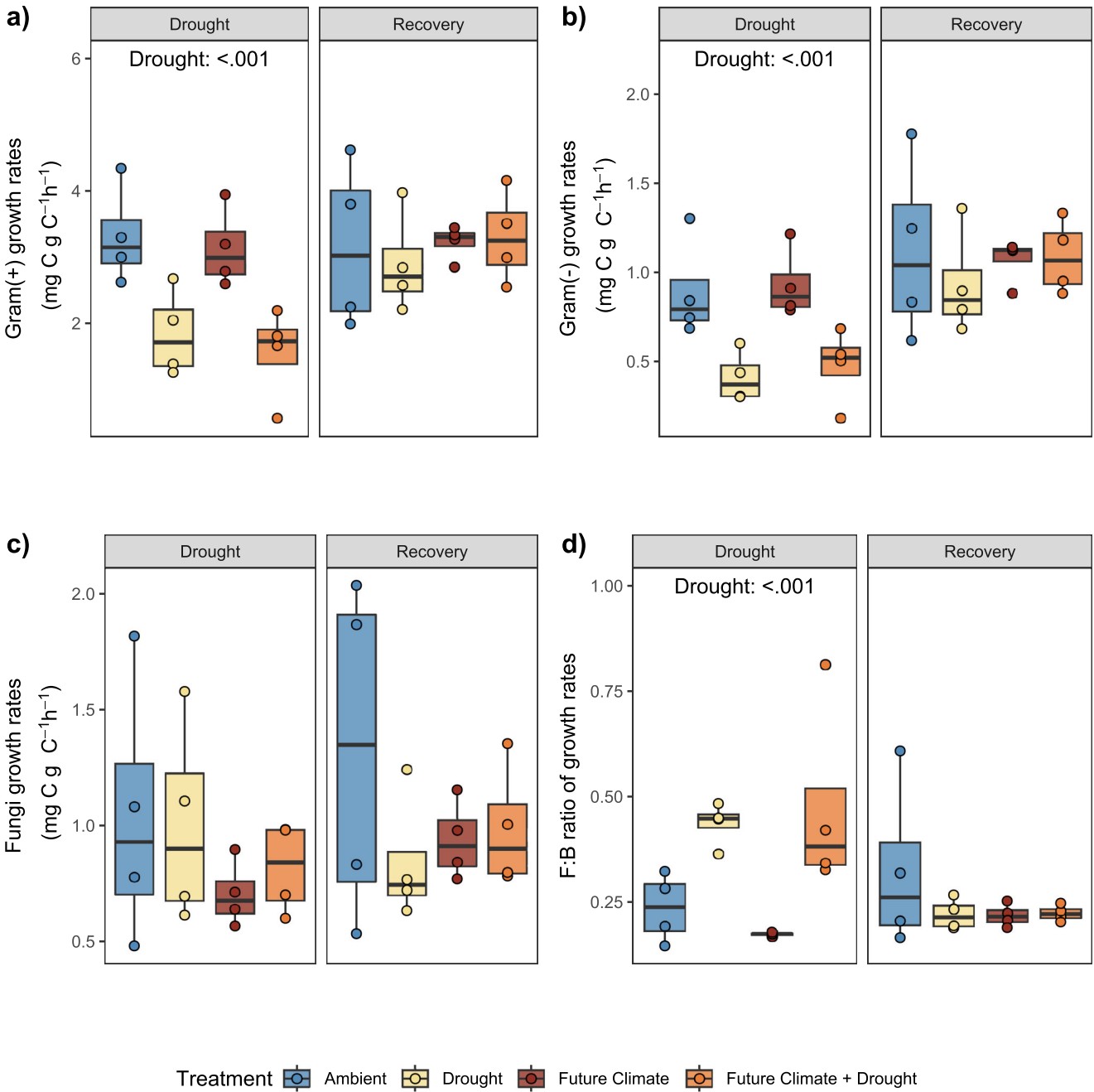

**Fig. 3 | Mass-specific growth rates of different microbial groups. a** Gram-positive and **b** Gram-negative bacterial markers and **c** fungal markers, as well as **d**) the ratio of fungal to bacterial growth rates at peak drought and recovery ('Drought' and 'Recovery'). Statistical results are reported for significant *p*-values (for a full report see Supplementary Table 4). Box centre line represents median, box limits the upper and lower quartiles, whiskers the 1.5x interquartile range, while separated points represent potential outliers. The sample size 'n' represents biologically independent samples (*n* = 4 replicates). (*n* = 4 replicates). Colour indicates treatment. Source data are provided with this paper.

was still remarkably resistant, remaining active and producing PLFAs even after an intense drought (two months of rain exclusion). Moreover, the microbial community showed a fast recovery after the drought ended. These empirical results are extremely important, as data from drought field studies are largely unavailable and sustained microbial activity during drought and recovery conditions could have large effects on soil C storage[18,49]. Finally, in contrast to our hypothesis, future climate conditions neither buffered nor enforced drought and recovery effects on microbial community level growth. While elevated $CO_2$ and temperature can stimulate microbial growth[46], our results demonstrate the overriding effect of drought over other climate change factors.

Fostering a mechanistic understanding of soil C decomposition rates and soil functioning, requires innovative approaches to quantitatively link microbial community composition to soil processes[50–52]. The use ²H-vapor-FAME-SIP to estimate phospholipid fatty acid production rates allows broad taxonomic groups to be separated across microbial kingdoms. Our results showed that the active community significantly differed in their composition between ambient conditions and future climate, as well as drought and rewetting (Fig. 2; Supplementary Table 3). Our approach also demonstrates that the abundance of PLFA markers does not clearly describe the active community at a certain time point. This discrepancy between growth- and biomass-based assessment highlights the importance of capturing growth rates

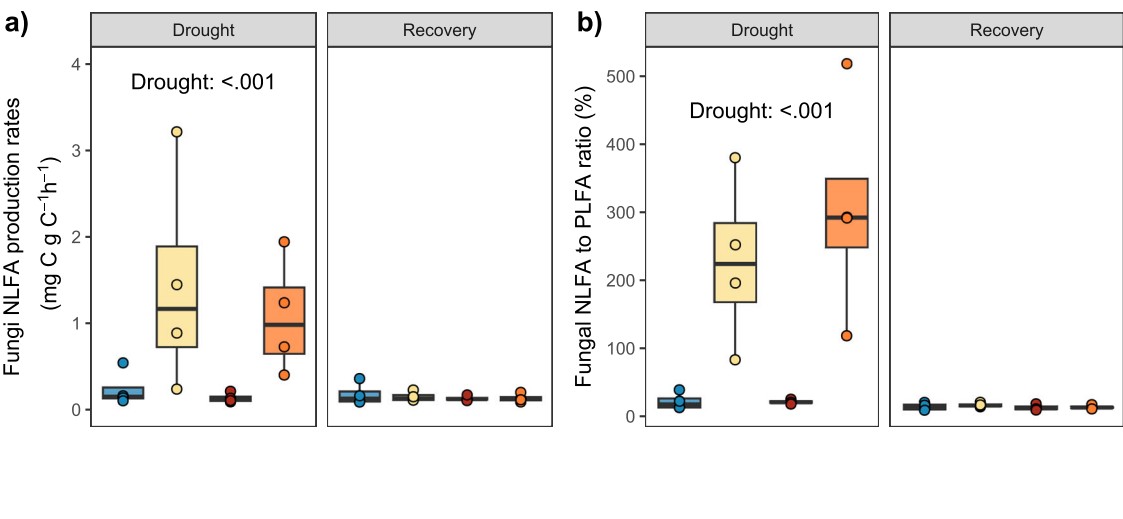

**Fig. 4 | Fungal investment into storage compounds. a** Production of fungal specific NLFA during 'Drought' and 'Recovery' periods. **b** Ratio of fungal specific newly produced NLFA to newly produced PLFA (expressed as percentage), indicating that fungi increase the relative investment in NLFAs during drought. This ratio allows to account for a potential underestimation of mass-specific NLFA production rates caused by potential necromass-NLFA accumulation. Statistical results are reported for $p$-values < 0.05 (for a full report see Supplementary Table 5). Box centre line represents median, box limits the upper and lower quartiles, whiskers the 1.5x interquartile range, separated points indicate potential outliers. The sample size represents biologically independent samples ($n$ = 4 replicates). Colour indicates treatment. Source data are provided with this paper.

with SIP approaches to understand microbial responses to environmental perturbations[53,54]. The growth of different soil microbial groups (fungi, gram negative and gram positive) strongly differed in their sensitivity to drought but was not affected by future climate conditions. Interestingly, both gram-positive and gram-negative bacteria similarly reduced growth during drought (Fig. 3). Although gram positive were suggested to be drought resistant because of their cell wall structure[55,56], only biomarkers specific for Actinobacteria displayed the least decrease in growth. This is in accordance to previous studies indicating Actinobacteria as drought resistant[28,48,57]. Fungal growth instead showed an astonishing resistance to drought and exhibited unchanged growth rates after a two-month summer drought in the field. While fungi have often found to be more resistant in their community composition[58], based on biomass and DNA measurements[15,58–62] and accessing labile plant C in the rhizosphere during drought[55], to our knowledge this is the first study directly demonstrating that soil fungal growth in soil displays complete resistant to drought in a field study. Fungi can resist drought via their extensive filamentous-like body structure (i.e., hyphae) that allows water and nutrients to be reached even when drought reduces diffusivity[15,56,63]. However, the lack of experimental approaches to eliminate potential biases caused by substrate and/or water addition has so far hindered the precise assessment of growth dynamics during drought conditions. Finally, our study also shows that bacterial growth was resilient and that all groups showed a fast recovery after drought.

Microbial biomass growth is classically supposed to be directly represented by cell replication rates. However, microorganisms can allocate large amounts of C to other processes, such as the formation of storage compounds, which can represent a large fraction of the total biomass[30]. The $^2$H-vapor-FAME-SIP method enabled us to additionally characterize the production rates of NLFAs (triglycerides), which represent major storage compounds in eukaryotic cells[64] and also in many bacterial species[39,65]. In fungal cells triglycerides can account for up to 80% of the cell dry weight, in extreme situations[66], but their ecological role has not received much attention[41,65]. Our results showed a large increase in storage compound synthesis during drought conditions, representing on average a 5.8 (drought) and 4.8-

fold (future climate + drought) increase compared to non-drought conditions. The production of NLFAs associated with fungi displayed the highest increase relative to the respective ambient conditions, which also led to an increase in the total pool of fungal NLFA (absolute amounts). Under non-drought conditions fungal storage compounds represented only around 22% of the investment in biomass (estimated by using the same PLFA biomarkers), which increased on average to 228% and 305% in drought and future climate + drought conditions, respectively, but production rates decreased fast during recovery and were indistinguishable from ambient controls. For arbuscular mycorrhiza it was shown that spores contain much more NLFAs than PLFAs, and NLFAs make up about 20% of the total spore biomass[67]. Hence, the fungal NLFA production observed in our study could also be an indication of spore formation and represent an important survival strategy during extended drought periods. Such fungal investments into storage enhance their resistance to stress conditions, which cannot be captured by using DNA-based methods. While information on PLFA turnover in soil is available, to our knowledge rates of NLFA degradation in soil are not known. Assuming a slower turnover than PLFA after cell death, our mass-specific estimates may underestimate NLFA production. Similarly, PLFA degradation could cause some newly formed fatty acids in PLFA to be measured within the NLFA pool after cell death and phosphate head removal. However, this contribution would be minimal given the short incubation time. Moreover, $^2$H-labelling could also be used to measure production rates of other storage compounds during drought, such as polyhydroxybutyrate (PHB), which represents an important storage compound of bacteria[65].

Drought can have negative impacts on the soil C balance[16], especially if soil microorganisms can maintain higher respiration activity compared to primary productivity[18,49]. Moreover, the increase in respiration and growth following rewetting of dry soils ('Birch effect'[68];) can lead to a decoupling of microbial anabolic and catabolic processes, and to more C being used for respiration[43]. Interestingly, in our study, microbial community-level CUE, describing the partitioning of C directed to growth versus respiration[63,69], remained unaffected by drought and also in the recovery period (measured five days after rewetting to avoid immediate respiration responses to rewetting right

after water additions[53]. This is in line with a previous study by Herron et al.[44], where microbial CUE (determined by $^{13}$C-acetic acid vapor addition in a laboratory incubation) decreased only at very low moisture levels below 0.05 g $H_2O$ g$^{-1}$ soil or −6.0 MPa; comparatively, drought treatments in our experiment reached on average 0.07 g $H_2O$ g$^{-1}$ soil (Supplementary Table 6). In contrast, we found a significant decrease in CUE by 43% ($^2$H-vapor-FAME-SIP) and 38% ($^{18}$O-vapor-DNA-SIP) caused by the future climate scenario (the combination of eCO$_2$ and warming), only during the recovery, similarly as in a previous study on the same site[45]. Other studies also showed that CUE can decrease with warming[70]. However, we previously showed that the response to eCO$_2$ and warming displayed nonlinear behavior and was strongly affected by seasonality[45], therefore these results should be interpreted with caution. Furthermore, in our experimental approach we used field temperature differences to represent conditions as close to the field as possible, and therefore measured growth rates were also affected by these conditions and not only by previous exposure to the climate change treatment.

We targeted two processes linked to cell replication: fatty acid and DNA production. Community-level rates of $^2$H incorporation into phospholipid fatty acids correlated with $^{18}$O incorporation into DNA and showed the same responses to drought and future climate conditions (Supplementary Fig. 1). However, by using $^2$H incorporation into fatty acids, the calculated community level growth rate and CUE were lower compared to DNA based estimates. Specifically, per unit of PLFA fewer new PLFAs were produced compared to the relative amount of newly formed DNA. Fatty acid $^2$H values reflect the isotopic composition of the source water[71]. However the amount of $^2$H derived from added labelled water can be considered as a combination of two factors: the mole fraction of water derived H that can be incorporated into fatty acids and the associated net $^2$H isotope fraction[72]. A previous study indicated that this can lead to as much as a 5-fold underestimation of growth rate if unaccounted for ref. 35. Therefore, to more accurately estimate microbial fatty acid production the physiological parameter 'water hydrogen assimilation constant (a$_w$)' should be applied[72,73], obtained from the regression of the $^2$H isotopic composition of fatty acids and water source. In a recent survey it was demonstrated that a$_w$ can range from values close to 0.1 up to values close to 1[74]. For complex microbial communities, such as in soil, the assimilation constant is difficult to estimate; but assuming all microbes are heterotrophs, this value was suggested to be on average 0.71[33], which we also applied in this study. However, it is important to note that using a different a$_w$ factor could lead to strong changes in the final quantification. For example, if we had used a value of 0.3, we would have generated almost identical PLFA and DNA- based growth rate estimates. An assimilation constant is currently not used for DNA, making the comparison between the two methods difficult. Nevertheless, relative differences between treatments in our experiment are independent of the chosen value and the uncertainty associated with the a$_w$ factor does not affect our conclusions.

Understanding terrestrial ecosystem responses to climate change events, such as drought, requires innovative approaches to quantify microbial growth and soil C cycling. Using $^2$H-vapor-FAME-SIP allows the concomitant measurement of community-level physiology and the direct quantification of growth and storage product synthesis of different microbial groups. Our results demonstrate that drought exerts a major control on microbial catabolic and anabolic processes. Despite the strong effects on microbial growth and respiration, soil microbial communities are able to maintain the balance between these processes, with surprisingly stable microbial CUE values both during drought and after rewetting in the recovery period. Most notably, we could unequivocally demonstrate that soil fungi remained active after a two-month field drought event. During drought, fungi invest large amounts of C into intracellular storage compounds, a strategy that could be connected to fungal resistance and resilience to drought.

Accounting for group specific growth rates and different physiological strategies of the growing soil microbes during global change, i.e., cell division versus storage compound synthesis, can foster our understanding of soil microbial contributions to the C cycle and ecosystem feedback on the climate system.

## Methods

### Experimental design
In a multifactorial global change experiment, we jointly manipulated atmospheric CO$_2$ concentrations (+300 ppm above ambient) and temperature (+3 °C above ambient; termed 'future climate condition' hereafter) and investigated how this condition altered the effects of drought compared to ambient controls (ambient, drought, future climate, future climate + drought).

### Site description, experimental layout and sample collection
Samples were collected from a managed montane grassland in the Austrian Alps, Styria, Austria (47°29′38″N, 14°06′03″E) as part of a multifactorial climate change experiment ('ClimGrass') located at the Agricultural Research and Education Center (AREC) in Raumberg-Gumpenstein, complying all ethical regulations of the AREC. The site is characterized by a mean annual temperature of 8.5 °C and a mean annual precipitation of 1077 mm. According to the WRB-system[75] the soil is classified as Dystric Cambisol (arenic, humic) with a loamy sand texture and a pH-value of ~5.5. Before establishment of the multifactorial global change experiment (ClimGrass), a typical grassland mixture was sown in an area of homogeneous soils in 2007, comprising the grass species *Arrhenatherum elatius* L., *Dactylis glomerata* L., *Poa pratensis* L., *Alopecurus pratensis* L., *Festuca rubra* L., *Trisetum flavescens* L., *Lolium perenne* L., *Phleum pratense* L. and *Festuca pratensis* L., and the legumes *Lotus corniculatus* L. and *Trifolium repens* L. The ClimGrass project entails 54 plots with a combined warming and Free-Air-Carbon dioxide-Enrichment (T-FACE) setup, put into full operation in 2014 to manipulate temperature and CO$_2$ at three levels each[45,76,77]. Fully automated rainout shelters were installed above half of the ambient and above half of the combined +3.0 °C and +300 ppm CO$_2$ (i.e.,'future climate') plots. All plots are harvested (plant biomass) three times a year (spring, summer and autumn) and receive identical rates of mineral fertilizer, applied in three batches giving a total load of 90 kg N, 65 kg P and 170 kg K per hectare and year.

For this experiment, we selected 16 plots representing four different treatments in a full factorial design ($n = 4$ per treatment, respectively): ambient ('ambient'), drought ('drought'), eCO$_2$ and elevated temperature combined (+300 ppm +3 °C; 'future climate'), and future climate with drought ('future climate + drought'). The drought period was simulated in the field between June 17th 2020 until August 3rd 2020 by excluding all naturally occurring precipitation. The drought plots then received a scheduled rewetting with 40 mm of previously collected rainwater on August 3rd 2020, after which the automatic rain-out shelters were switched off and the plots were used to investigate the recovery from drought. The volumetric soil water content (SWC) was measured at 1-min intervals at 3- and 9 cm depth with soil moisture sensors (SM150T, DeltaT) in a subset of representative plots throughout the growing season. Values are displayed as daily averages (Supplementary Fig. 1). Average soil moisture values measured gravimetrically at the time of sampling are reported in Supplementary Table 1.

We collected soil samples in two campaigns, towards the end of the drought period (29th of July) and two days after the rewetting event (5th of August). Hereafter, we refer with 'Drought' to the samples that were collected at the end of the severe drought period, and with 'Recovery' to the soil samples that were collected after rewetting. From each experimental plot, three soil samples were collected using a soil corer with a diameter of 2 cm and a length of 10 cm. For each plot, samples were pooled, fresh masses weighed, sieved to 2 mm and fine

roots were removed. Aliquots of fresh sieved soil were weighed and dried (105 °C, 48 h) to calculate soil water content. Further fresh soil aliquots were used for determining microbial biomass based on the chloroform fumigation extraction method described by[78], as well as setting up the laboratory incubations (see below).

### Laboratory incubation set-up: soil microbial activity using $^2$H-vapor-SIP compared to $^{18}$O-vapor-SIP

We tested the applicability of $^2$H incorporation into phospho- and neutral lipid fatty acids (PLFA and NLFA) to determine soil microbial group specific growth and buildup of storage compounds under drought and future climate change conditions. Although the extracted PLFA fraction may contain other lipid classes[79], representing minor component of membrane lipids, we use the term PLFA throughout the manuscript as this is the major fraction analysed. For labelling soil water with $^2$H we used the water vapor equilibration method as described by[27] for $^{18}$O incorporation into DNA. For clarity, we will refer to the $^2$H-tracing approach as $^2$H-vapor-SIP. In addition, we compared the PLFA-based ($^2$H-vapor-SIP) to DNA-based ($^{18}$O-vapor-SIP) estimates for microbial growth and C use efficiency ($CUE_{PLFA}$ vs. $CUE_{DNA}$). The short-term laboratory incubations were started within 24 h after sample collection.

We used a similar experimental set up as described in detail in[27], but adjusted soil sample size based on the amount of soil needed for extracting sufficient amounts of fatty acids or DNA, respectively: we incubated around 800 mg (in duplicates) to trace $^2$H-labelled water into PLFAs and NLFAs and around 400 mg of fresh soil to trace $^{18}$O labelled water into DNA. Soil samples were weighed into 1.2 ml plastic vials and inserted in 27 ml glass headspace vials, which were closed airtight with rubber septa. For both assays ($^2$H-vapor-SIP and $^{18}$O-vapor-SIP) the labelled water was applied at the bottom of the glass headspace vial with no direct contact to the soil. The amount of water added was calculated based on the amount of water that would increase the soil water to 60% of their respective water holding capacity to aim for an approximate enrichment of the soil water of around 20 at% $^2$H and 20 at% $^{18}$O, by the end of the incubation.

Samples were incubated at their respective field temperatures at the time of sample collection (ambient, drought: 20 °C; future climate, future climate + drought: 23 °C) for 48 h. For both assays $CO_2$ concentrations in the glass headspace vials were determined at the beginning and end (after 48 h) of the incubation to calculate soil respiration rates (see details below). At the end of the incubation the soil samples were removed from the headspace vials, closed, and shock frozen in liquid $N_2$, then kept at −20 °C until further analyses.

### Temporal dynamics of $^2$H and $^{18}$O equilibration with soil water

At the 'Drought' collection we randomly selected several samples (two for each treatment) for both $^2$H and $^{18}$O incubations. From this we collected aliquots of the remaining isotopically labelled water ($^2$H or $^{18}$O) from the bottom of the vial after 3, 6 and 16 h of incubation. In addition, we collected the water at the end of the 48 h incubation, to calculate the incorporation of $^2$H and $^{18}$O into soil water as described in Canarini et al. (2020; Supplementary Fig. 11 and Supplementary Fig. 12). The $^2$H and $^{18}$O enrichment of the soil water is used to calculate PLFA and DNA production and ultimately microbial growth, and the average $^2$H and $^{18}$O enrichment of soil water needs to be calculated across the incubation time (48 h) to account for the temporal dynamics of isotope equilibration of soil water. To do so, we used the same approach as described in Canarini et al.[27]. Briefly, we collected the water left at the bottom of the headspace vials in all incubated samples at the end of the incubation period (48 h); additionally a subset of samples ($n = 2$ per treatment) were set up similarly as described in the Methods section to collect the water several times (after 3, 6 and 16 h) across the incubation period. The collected water samples were analyzed for $^2$H and $^{18}$O enrichment, respectively. The

resulting curves represent the equilibration of the $^2$H and $^{18}$O labelled external water via the vapor phase with soil water, and were fitted with a negative exponential function (Eq. 1) as described in Ingraham and Criss (1993):

$$^2H \, at\%_{soil\,water} = {}^2H \, at\%_{48} + ({}^2H \, at\%_{in} - {}^2H \, at\%_{48}) * (e^{-bt}) \qquad (1)$$

where $^2$H at$\%_{48}$ and $^2$H at$\%_{in}$ represent the $^2$H atom % (or $^{18}$O) of the water after 48 h incubation and at time point 0, and $b$ represents a soil specific coefficient that was generated by fitting the nls() function in R. We used the values ($^2$H at$\%_{48}$ and $b$) to generate a prediction of the soil water isotopic enrichment as was described for $^{18}$O demonstrated in Canarini et al.[27]. Finally we calculated the integral of this function by using the function *integrate()* of the R package "pracma" between time 0 and 48 h. This integral was divided by 48 h to generate an average isotopic enrichment (the term at$\%_{soil\,water}$ in Eq. 1, and the term $^2$H at $\%_{soil\,water}$ in Eq. 2 described below) of soil water for each treatment (values from ambient samples were used for the 'Recovery' period).

The collected $^2$H-labelled water was analyzed using platinum catalyzed equilibration of $^2$H in H$_2$O with H$_2$ gas by a Gasbench II headspace sampler connected to a Delta V Advantage isotope ratio mass spectrometer (Thermo Fisher, Bremen, Germany). The sampled $^{18}$O-labelled water was analyzed through equilibration of $^{18}$O in H$_2$O with $CO_2$ by a Gasbench II headspace sampler connected to a Delta V Advantage isotope ratio mass spectrometer (Thermo Fisher, Bremen, Germany). Both $^2$H and $^{18}$O values of the collected water samples were calibrated against isotope calibration curves using water with known isotopic values. We calculated the $^{18}$O and $^2$H soil water enrichment via measuring loss of enriched isotope in the added source water (Canarini et al.[27]). The $^2$H incorporation showed slower equilibration curves but reached comparable isotopic enrichment to the $^{18}$O method after 48 h (Supplementary Fig. 11 and Supplementary Fig. 12). Average values of soil water enrichment (expressed in at%) for $^2$H during the 'Drought' sampling were 16.1 for ambient, 12.6 for drought, 18.9 for future climate and 15.5 for future climate + drought. For the incubation of samples collected in the 'Recovery' period the $^2$H enrichment was 16.1 for ambient and drought, and 18.9 for future climate and future climate + drought. Average values of soil enrichment (expressed in at %) for $^{18}$O during the 'Drought' sampling were 20.3 for ambient, 18.9 for drought, 21.5 for future climate and 19.9 for future climate + drought; for the incubations conducted in the 'Recovery' period $^{18}$O enrichment of soil water was 21.9 for ambient, 20.6 for drought, 22.7 for future climate and 21.3 for future climate + drought. Additionally, we verified that $^2$H does not inhibit microbial activity. We measured respiration in both labelled and natural abundance samples and tested with a pair $t$-test if respiration rates were altered by the introduced $^2$H label. Results show no significant difference (t = 1.992; $p = 0.0534$; Supplementary Fig. 13a) with a small deviation for high values from the 1:1 line of respiration rates of labelled and natural abundance samples (Supplementary Fig. 13b).

### Microbial growth, respiration and CUE determined via $^2$H-vapor-SIP

Soil microbial growth, microbial respiration and microbial C use efficiency ($CUE_{PFLA}$) were determined based on the incorporation of $^2$H from soil water into PLFAs. Additionally, we also calculated investment into lipid storage compounds via the incorporation of $^2$H from soil water into NLFAs (extracted as described below). In fatty acids the hydrocarbon skeleton consists of non-exchangeable C-H bonds which allows $^2$H-incorporation to be used as an indicator for biological activity[80]. Fatty acid biosynthesis and subsequently also $^2$H incorporation combine fatty acid production related to membrane growth, but also membrane repair; therefore, calculated growth rates need to be considered accordingly. Microbial respiration was determined by measuring the $CO_2$ concentration in the headspace vial right after the

application of $^2$H enriched water and 48 h after the incubation using an infrared gas analyzer (EGM4, PP systems).

PLFAs and NLFAs were extracted from freeze-dried soil samples with a modified high throughput method[39]. Total lipids were extracted from around 1 g of soil using a chloroform:methanol:citric acid buffer mixture and fractionated by solid-phase extraction on silica columns. The neutral lipid fatty acid (NLFA) fraction was collected by eluting samples with chloroform (containing 2% ethanol as recommended in[81], subsequently, the PLFA fraction was collected by eluting columns with a 5:5:1 chloroform:methanol:water mixture. After an internal standard (fatty acid methyl ester: 19:0) was added, NLFAs and PLFAs were converted to fatty acid methyl esters (FAMEs) by transesterification. This method does not derivatize free fatty acids, which are therefore not included in the analysis[82]. The internal standard is added as FAME and therefore our approach does not fully account for extraction or derivatization efficiency. To minimize differences between samples, all the vials were derivatized in the same day under the same conditions. Samples were analyzed for H quantity and $^2$H incorporation using a Trace GC Ultra connected by a GC-IsoLink to a Delta V Advantage Mass Spectrometer (all Thermo Fisher Scientific). Samples were injected in splitless mode (injector temperature 300 °C) and separated on a DB5 column (60 m × 0.25 mm × 0.25 µm; Agilent, Vienna, Austria) with 1.2 ml He min$^{-1}$ as the carrier gas (GC program: 1 min at 80 °C, 30 °C min$^{-1}$ to 150 °C, 1 min at 150 °C, 4 °C min$^{-1}$ to 230 °C, 30 min at 230 °C, 30 °C min$^{-1}$ to 280 °C and 19 min at 280 °C). A subset of samples from each treatment were used for peak identification and injected on a GC (7890 B Agilent Technologies, USA) connected to a time of flight (TOF) MS (Pegasus BT, LECO, USA). The instrument was equipped with a split/splitless injector and samples were run with the same column and program as the GC-IRMS, to retain the same peak order. For identification of FAMEs we used two external mixed standards, i.e. the fatty acid methyl ester mix (FAME mix, Supelco, USA), and the bacterial fatty acid methyl ester mix (BAME mix, Supelco, USA). Unknown FAME were identified via mass spectra comparisons with the NIST libraries *mainlib* and *replib*, as well as via comparison with a personal library developed via pure cultures of soil bacteria and fungi[39], and quantified against the internal standard (19:0). The H3 factor (correction for abundance of the trihydrogen ion H$_3^+$) was defined based on multiple injections of reference hydrogen gas before each sequence, and was stable over time (3.64 ± 0.05 ppm/nA). Sample δ$^2$H values were normalized to the VSMOW scale using the slope and intercept of measured and known values of isotopic standards. These were individual and fatty acid mixes of USGS70 and USGS72 (Arndt Schimmelmann, Indiana University) measured before and after each measurement run, as described above, and ranging from δ$^2$H −183‰ to 384‰. Offsets between measured and known values for these standards were used to correct for any drift over the course of the sequence or any isotope effects associated with peak area or retention time. The standard deviation for these standards averaged 4‰ and the average offset from their known values was 27‰. Memory effect corrections were not applied; the maximal potential offset ranges around 4%[83] and analysis of methane reference standards that follow analytical peaks suggested that memory effect should be at around 0.2%[73]. For $^2$H we obtained an average biomarker isotopic value across labelled samples of 0.207 at% for PLFA (natural abundance value averaged at 0.0136; Supplementary Fig. 14) and of 0.0609 at% for NLFA (natural abundance = 0.014; Supplementary Data 1).

We used the markers 18:1 ω9cis and 18:2 ω6,9 to estimate fungal biomass[84,85]. For actinobacteria markers we used the sum of 10Me16:0, 10Me17:0 and 10Me18:0. Gram-positive bacteria were calculated by the sum of actinobacteria biomarkers plus the i15:0, a15:0, i16:0, i17:0 and a17:0 markers, for gram-negative bacteria we used the markers 16:1 ω7, cy17:0 and cy19:0[86]. Fungi to bacteria ratio was calculated as the ratio between the fungal biomass and the sum of gram positive and gram negative. We used the markers

16:1 ω5 as indicative of arbuscular mycorrhizae fungi, but kept separated from the other fungal biomarkers (except for the multivariate analysis). The remaining markers including 18:1 ω9trans, 16:0, 17:0, and 18:0 cannot be exclusively assigned to bacterial or fungi and markers with double bond position which could not be chromatographically resolved, were assigned to the general PLFA marker group[86].

We calculated the newly produced C in each fatty acid (FA C$_{produced}$) and expressed as ng C g$^{-1}$ soil dry weight, using the enrichment of $^2$H as:

$$FA\ C_{produced} = \frac{(2H\ at\%_{labelled\ FA} - 2H\ at\%_{nat.ab\ FA})}{a_w \times 2H\ at\%_{soil\ water}} \times FAC \quad (2)$$

with $^2$H at%$_{labelled\ FA}$ describing the atom% $^2$H of a specific fatty acid, $^2$H at%$_{nat.ab.\ FA}$ representing the natural $^2$H abundance of the same fatty acid (sample taken from the same plot and incubated with natural abundance water), at%$_{soil\ water}$ indicating the $^2$H atom% in the soil water estimated by the equilibration curves described in above. The factor $a_w$ represents the assimilation efficiency of deuterium into fatty acids and it has been estimated at a value of 0.71 for heterotrophs[33,72]. 'FA C' is the C concentration of the respective fatty acid (average of labelled and natural abundance sample) calculated relative to the known amount of C of the internal standard 19:0.

For each sample we calculated the total soil microbial community C production as the sum of all measured FA C$_{produced}$ for PLFAs (PLFA C$_{produced}$) or NLFAs (NLFA C$_{produced}$), respectively. PLFA C$_{produced}$ can be used to calculate microbial biomass growth rates (Growth$_{PLFA}$ in ng C g$^{-1}$ dw soil h$^{-1}$), using soil microbial biomass C determined by chloroform fumigation extraction[78] as follows:

$$Growth_{PLFA} = \left(\frac{PLFA\ C_{produced}}{Total\ PLFA\ C} * Microbial\ BM\ C\right) / time \quad (3)$$

by multiplying the fraction of PLFA C$_{produced}$ per total PLFA C (i.e., the sum of all FA C per sample, Total PLFA C) with the amount of microbial biomass C (Microbial BM C in µg g$^{-1}$ dw soil) of each soil and divided by the incubation time. The amount of C taken up by the microbial community (C$_{Uptake\ PLFA}$) was estimated as

$$C_{Uptake\ PLFA} = Growth_{PLFA} + C_{Respiration} \quad (4)$$

where Growth$_{PLFA}$ is the C allocated to biomass production (ng C g$^{-1}$ dw soil h$^{-1}$), and C$_{Respiration}$ is the C allocated to the production of CO$_2$ (ng C g$^{-1}$ dw soil h$^{-1}$). In addition, we calculated mass-specific growth rates for the Growth$_{PLFA}$ (PLFA-MS Growth rates) by dividing the values by Microbial BM C (final unit: mg C g$^{-1}$ mic C h$^{-1}$) as well as by multiplying by 24 (final unit: day$^{-1}$), and also for each microbial group separately, by summing up FA-C$_{produced}$ for each group using the assignment of FA described above (for both PLFA and NLFA separately), divided by the respective FA C and incubation time (mg C g$^{-1}$ C h$^{-1}$). For NLFA we also calculated the percentage of NLFA biomarker produced relative to the same PLFA biomarker and grouped by microbial group as per the assignment described above. Microbial CUE$_{PLFA}$ was then calculated by the following equation[63,69]:

$$CUE_{PLFA} = \frac{Growth_{PLFA}}{Growth_{PLFA} + C_{Respiration}} \quad (5)$$

And microbial community turnover time T (expressed in days) was calculated as:

$$T = \frac{Microbial\ BM\ C}{Growth_{PLFA}} \times 24 \quad (6)$$

## Microbial growth respiration and CUE determined via $^{18}O$-vapor-SIP

As comparison to PLFA based microbial growth and CUE, we also determined DNA based growth (Growth$_{DNA}$) and C use efficiency (CUE$_{DNA}$) based on the incorporation of $^{18}O$ from soil water into genomic DNA via $^{18}O$-vapor-SIP as described earlier[27]. Microbial respiration was determined by measuring the $CO_2$ concentration in the glass headspace vial as described above. DNA was extracted using a DNA extraction kit (FastDNA™ SPIN Kit for Soil, MP Biomedicals). DNA concentration of each extract was determined fluorimetrically following the Picogreen assay (Quant-iT™ PicoGreen® dsDNA Reagent, Life Technologies). Subsequently, the $^{18}O$ enrichment and the total O content of the purified DNA fractions were measured using a Thermochemical elemental analyzer (TC/EA Thermo Fisher) coupled via a Conflo III open split system (Thermo Fisher) to an isotope ratio mass spectrometer (Delta V Advantage, Thermo Fisher). We obtained an average isotopic value across labelled samples of 0.214 at% (natural abundance value averaged at 0.203 at%; Supplementary Data 1). The amount of DNA produced over the 48 h of incubation can be calculated using the following formula as also described in Canarini et al.[27]:

$$DNA_{produced} = O_{DNA\ extr} * \frac{^{18}O\ at\%_{labelled\ DNA} - {}^{18}O\ at\%_{n.a.\ DNA}}{^{18}O\ at\%_{soil\ water}} * \frac{100}{31.21} \quad (7)$$

where $O_{DNA\ extr}$ is the total amount of oxygen in the DNA extract, $^{18}O$ at%$_{DNA\ L}$ and $^{18}O$ at%$_{DNA\ n.a.}$ are the $^{18}O$ enrichments of labeled and unlabeled DNA extracts, respectively, $^{18}O$ at%$_{soil\ water}$ is the $^{18}O$ enrichment of the soil water (calculated as average $^{18}O$ enrichment in soil water from the equilibration curves described in above), multiplied by the fraction of the average oxygen content in DNA (31.21%). We here used the amount of DNA$_{produced}$ (µg g$^{-1}$ soil dry weight) to calculate microbial growth in units of C by multiplying it with the ratio of microbial biomass C (determined by chloroform fumigation extraction[78] to DNA content of each soil sample:

$$Growth_{DNA} = \frac{DNA_{produced}}{Total\ DNA} * Microbial\ BM\ C \quad (8)$$

Similarly, the amount of C taken up by the microbial community based on $^{18}O$-vapor-SIP into DNA (C$_{Uptake}$) is estimated as:

$$C_{Uptake\ DNA} = Growth_{DNA} + C_{Respiration} \quad (9)$$

where Growth$_{DNA}$ is the C allocated to biomass production, and C$_{Respiration}$ is the C allocated to the production of $CO_2$. In addition, we calculated mass-specific growth rates for the Growth$_{DNA}$ (DNA-MS Growth rates) by dividing the values by Microbial BM C (final unit: mg C g$^{-1}$ mic C h$^{-1}$) as well as by multiplying by 24 (final unit: day$^{-1}$). Microbial CUE$_{DNA}$ was then calculated as described above:

$$CUE_{DNA} = \frac{Growth_{DNA}}{Growth_{DNA} + C_{Respiration}} \quad (10)$$

And microbial community turnover time T (expressed in days) was calculated as:

$$T = \frac{Microbial\ BM\ C}{Growth_{DNA}} \times 24 \quad (11)$$

## Statistical analysis

Statistical differences in community level mass specific growth rates, microbial abundance, CUE and NLFA production rates were assessed by a linear mixed effect model separated for peak drought and rewetting sampling, using the package 'nlme'[87]. In all the models we tested for the effects of drought, future climate and their interaction. Plot number was used as a random factor to account for plot variability. Differences in respiration between samples where $^{2}H$ was added compared to natural abundance water were evaluated via two-sided paired t-test, via the R function 't.test'. Correlations between CUE and mass specific growth for the method comparison were assessed via Pearson's correlation via the 'stat_cor' function. Model assumptions were inspected visually, and values were log-transformed when assumptions were not met. Principal component analyses of relative PLFA abundances and growth rates were assessed via the function 'PCA' of the package 'FactoMineR'[88] and scores were normalized by setting scale = TRUE. Statistical significance of relative PLFA abundances and growth rates between treatments was assessed via PERMANOVA with the function 'adonis' of the 'vegan' package[89] using Euclidean distances. For both PCA and PERMANOVA data was split, with 'Drought' and 'Recovery' separately. Plots were generated via the package 'ggplot2'[90]. Statistical analyses were performed in R 3.6.3[91].

## Reporting summary

Further information on research design is available in the Nature Portfolio Reporting Summary linked to this article.

## Data availability

Source data are provided as a Source Data file. All data used in this study are available in the Supplementary Information file and in Supplementary Data 1. Source data are provided with this paper.

## Code availability

The code used to generate figures and statistical analyses is provided at https://github.com/acanarini/Soil-fungi-drought with the following https://doi.org/10.5281/zenodo.14048057.

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

## Acknowledgements

We would like to thank the team of the Austrian Research and Education Centre Raumberg-Gumpenstein (HBLFA) for their support during the sampling campaign and for the provision of the experimental site, which was supported by the DaFNE project ClimGrassEco (101067), the FWF project P28572-B22, and funding of the Earth System Sciences research program of the Austrian Academy of Sciences (ÖAW project ClimGrassHydro) obtained by MB. LF acknowledges the European Union's Horizon 2020 research and innovation program under the Marie Sklodovska-Curie grant agreement no. 847693 (REWIRE). DM was financially supported by FutureArctic, a European Union's Horizon 2020

research and innovation program under the Marie Skłodowska-Curie Actions (grant no. 813114). This research was funded, in part, by the Austrian Science Fund (FWF), Cluster of Excellence COE7 (grant DOI 10.55776/COE7). For the purpose of open access, the author has applied a CC BY public copyright licence to any Author Accepted Manuscript version arising from this submission. We thank Ludwig Seidl for technical support with GC-IRMS measurements.

## Author contributions

Conceptualization: A.C., L.F., J.S., A.R. Methodology: A.C. L.F., J.S., M.W., D.N., A.K., A.R., Investigation: A.C., L.F., J.S., D.M., M.W. Resources: A.R., M.B., L.F., E.P., A.S. Visualization: A.C., L.F. Supervision: A.R., A.C., L.F., J.S., M.B. Writing—original draft: A.C., L.F. Writing—review & editing: All the authors.

## Competing interests

The authors declare no competing interests.

## Additional information

**Peer review** *Nature Communications* thanks the anonymous reviewers for their contribution to the peer review of this work. A peer review file is available.

