## [Peer Review file · Nature Communications]

Soil fungi remain active and invest in storage compounds during drought independent of future climate conditions.

Corresponding Author: Dr Alberto Canarini

Version 0:

Reviewer comments:

Reviewer #1

(Remarks to the Author)

The paper by Canarini et al. uses D2O labelling to investigate the synthesis of membrane and storage lipids of soil microbes. They made use of a field experiment with "future climate" and drought treatments. Hence the design allowed them to test how microbial ecophysiology is affected by drought and future climate. The main novelty of the work is application of D2O labeling and subsequent measurement of 2H incorporation into membrane and storage fatty acids. Critically, they used a vapour phase approach to label soil and thus, unlike liquid D2O addition, they managed to estimate synthesis without affecting soil water content. Their work revealed that fungi remained active and invested into storage compounds under the drought conditions.

The experimental work is thorough and data interpretation is fine, so most of my comments relate to use of literature and tightening concepts.

One of the main findings is that fungi are less affected by water deficits than bacteria, but interpretation would benefit from considering the severity of water deficit. I recommend bringing water deficit to the fore to help contextualise the differential responses of bacteria and fungi. Was the stress severe or mild? What sort of water potentials were involved?

In quite a few places it seemed PLFA was being used inappropriately. The first rather minor issue is that PLFA is a misnomer given that the fraction can contain fatty acids from lipid classes in addition to phospholipids. I don't think here is the place to introduce new terminology because PLFA is firmly entrenched in the literature. Nevertheless, in the interest of accuracy, avoid phospholipid and instead speak of membrane lipid. The more confusing second issue is that "PLFA" is in some places used to refer to fatty acids from storage lipids. This invites confusion and could mislead into thinking fatty acids of membrane and storage lipids were measured simultaneously.

Line 317-. Speaks of turnover of membrane lipids in soil, but seems to misinterpret the data. Available data show that turnover of phospholipids is very rapid after death or when added to soil (reference 54). However, the turnover of membrane lipids within microbes is far, far slower (ref 78). Likely due to C-H of lipids (and presumably much of the fatty acid backbone) being re-used multiple times.

It's a moot point if the comparison of DNA vs fatty acid production adds a lot to the manuscript. It's definitely an interesting methodological question, but additional data are required to tease apart the reasons for differences between methods. I say this because there are many unknowns and assumptions in both approaches. Without additional mechanistic insights the conclusion is a bit prosaic: the methods differ but we don't know why or which is more accurate. Ultimately the largest challenge is that recycling of C-H bonds of lipids means growth is not wholly dependent on lipid synthesis. Also, the discussion of factors affecting 2H enrichment may be conflating (or combining?) two different factors. First there is the fractionation that can occur between source water and C-H in a lipid. The independent second factor is the fraction of C-H bonds in a lipid that can be labelled.

Minor comments

Line 123. There are quite a few other studies that have used D2O labeling to investigate synthesis of lipids in soil: Huguet, A., Meador, T.B., Laggoun-Défarge, F., Köneke, M., Wu, W., Derenne, S., Hinrichs, K.-U., 2017. Production rates of bacterial tetraether lipids and fatty acids in peatland under varying oxygen concentrations. *Geochimica et Cosmochimica Acta* 203, 103-116.
Warren, C.R., 2023. LC-MS analysis of D2O-labelled soil suggests a large fraction of membrane lipid exists within slow growing microbes. *Soil Biology and Biochemistry* 177, 108912.
Wegener, G., Bausch, M., Holler, T., Thang, N.M., Prieto Mollar, X., Kellermann, M.Y., Hinrichs, K.U., Boetius, A., 2012. Assessing sub-seafloor microbial activity by combined stable isotope probing with deuterated water and ¹³C-bicarbonate. *Environ Microbiol* 14, 1517-1527.
Wegener, G., Kellermann, M.Y., Elvert, M., 2016. Tracking activity and function of microorganisms by stable isotope probing of membrane lipids. *Current Opinion in Biotechnology* 41, 43-52.

Line 36. Rather than “fatty acids”, I think there’s a need to specify membrane and storage lipids

Line 126. Not correct. PLFA-SIP can measure only membrane lipids. A separate (not concurrent and not PLFA) measurement is required for measuring production of storage compounds. Also, reference 38 does not contain data that support this point.

Line 134-135. Reference 38 does not contain empirical data so doesn’t really support this point.

Line 154, hypothesis 3. I think the logic needs explaining. Is the logic that water deficit reduces nutrient supply more than C and the increase in C:nutrient leads to storage? Or is the logic related to dormancy?

Fig 2 legend. It’s not clear to me what data were used for the PCA. E.g. Is it rate of ²H enrichment, or rate of synthesis?

Reviewer #2

(Remarks to the Author)

The work by Canarini et al. builds upon recent advances in ¹⁸O-DNA stable isotope probing and deuterium-based lipid stable isotope probing to measure the biomass production rates of fungal and bacterial lipid compounds in soil. The authors provide a valuable comparison between these methods to measure incorporation of ²H and ¹⁸O into lipid and DNA pools, respectively, which can relate the production rates of lipid and nucleic acid biomass, and their relationship to respiration rate. Furthermore, the authors examine the neutral lipid fraction to measure the production rates of storage compounds – namely FAs derived from triglycerides. All of these measurements are undertaken in the context of future climate conditions including elevated CO₂ (eCO₂) and drought. The authors include monitoring of microbial respiration processes in order to estimate carbon use efficiency (CUE) in their soil conditions. The positioning of this study in the scientific literature is important and the results of such a study would be of interest to soil microbiology community. I find the comparison of lipid-SIP and DNA-SIP to be very timely given heightened research in both of these areas within the past few years. The use of vapor-based tracers represents a further advancement towards more passive measurement of microbial activity in situ. The authors infer logical conclusions from their data regarding the effects of drought and future climate conditions on microbial growth and storage and I have no issues with their discussion or how it informs soil C dynamics more broadly.

Unfortunately, while the conclusions drawn on the data are valid, I have serious concerns regarding the data itself and the methodology, which does not meet the expected standards of the organic geochemistry field. In the paper’s current form, major sources of uncertainty remain unaddressed and, if unanswered, would leave the conclusions of the study unjustified. I cannot recommend this work for publication at Nature Comms without the authors undertaking substantial work addressing the following issues, which, at the discretion of the editor, could be outside the scope of the current submission.

My intention with writing this review is not to be overly harsh to the authors. Indeed, I was excited to see the preprint of this work as it represents the first study (to my knowledge) to co-register DNA- and lipid- SIP strategies, and certainly the first to do it with vapor equilibration. I identify deficiencies mainly in relation to the core lipid-SIP measurements of the study because these problems are glaring and would be quickly noted by microbiologists familiar with soil organic geochemistry. At the end of my review, I suggest potential ways the authors can reframe and improve the manuscript.

Major Comments

1. I am concerned that there is no reference to or description of the isotopic values of the PLFAs, NLFAs, and DNA held within the main text, supplementary texts, or SI data. This is critical context for readers to contextualize the extent of isotopic labelling that has occurred. I would recommend a figure, either in main text or supplement that shows, for each incubation condition, differentiating specific compounds (specific PLFAs, NLFAs, and DNA), the observed increase in ²H and/or ¹⁸O over time. Including raw data as a supplementary table would also be useful for reproducibility. The natural abundance isotopic values of these compounds in your study site would need to be included, as well as the effective isotopic label strength the samples experienced. This is important to include for multiple reasons. First, quantitative isotopic labeling occurs only when i) sufficient isotopic tracer is incorporated above background noise and ii) the extent of incorporation does not approach that of the label. Without this information, readers cannot assess the precision of your quantification. Second, because D₂O enrichments of > 20 at. % were applied to the soils, it is possible that extremely high ²H enrichments of

FAMEs were achieved. If this were the case, I would have further concerns regarding the quality of IRMS data (e.g., mass 3 and mass 2 peaks beginning to overlap, risk for chromatography shifts due to multiply-substituted FAMEs, memory effects, etc.), but without the data presented, neither I nor future readers can assess these factors. I consider this a major omission that must be addressed prior to publication.

2. Regarding NLFAs, it is unclear to me how the authors differentiate between triglyceride-sourced NLFAs (from storage compounds) and free fatty acids in the environment. Free fatty acids, sourced from both living and dead biomass from all domains of life, would certainly be present in these soils. During organic extraction and separation, free FAs would co-elute with FAs sourced from triglycerides in the NLFA fraction. In the methods section, no mention is made whether free FAs were quantified and distinguished from triglyceride-FAs. The conclusion that is most at risk of this problem is the finding that fungi produce NLFAs during drought. Free FAs in the environment can be sourced from microbial biomass as well as eukaryotic biomass (e.g., degraded plant tissue) and these free FA concentrations could vary widely between soil types and conditions. It also seems possible to me that PLFAs could have been produced (by fungi or others), these cells lyse, causing their PLFAs to hydrolyze to free FAs, and these free FAs eluting in the NLFA fraction would be mistakenly attributed to triglycerides. This risk may be high due to the rapid degradation of PLFAs (on the order of hours – see cited Zhang et al.).

3. Another point regarding NLFAs: the major assumption inherent to all PLFA studies is that PLFAs degrade rapidly upon cell death (the authors appropriately cite the recent Zhang et al. work on this). PLFA degradation enables their use as conservative tracers of microbial biomass production because it targets analyses at living biomass. I am unaware of any study measuring the triglyceride degradation rate in soil. (From a quick literature search on my end, it appears that these rates in soil are unconstrained). Due to a lack of a polar head group, one would expect that rates of triglyceride degradation would be far slower than those of PLFAs. Therefore, it seems fraught to present NLFA/triglyceride production rates without commenting on the strong possibility that a high concentration of NLFAs are sourced from necromass. If the measured NLFA pool contains substantial necromass, calculated rates of NLFA-related growth would not be quantitative, as they would be dependent on the amount of relict NLFA necromass in each soil condition. At the minimum, the authors would need to state upfront the caveat that their NLFA production rates are under-estimates due to the relict FAs present in the environment and the fact that triglycerides from dead cells are not rapidly lost from the NLFA pool.

4. I have major concerns regarding compound identification and ascription to microbial groups:

a. I am concerned that the authors' compound identifications may be incorrect, which would call into question their ability to distinguish microbial taxonomic groups. The authors state on lines 542 – 545: "We used the markers 18:1w9cis and 18:2w6,9 to estimate fungal biomass" and continue to state the markers used for gram+ and gram- bacteria. The authors identify these compounds using the retention times of BAME-CP and 37FAME standard mixtures (line 530). The issue here is that a GC-IRMS does not provide structural information regarding the unsaturation of the FAME compounds. For example, an 18:1w9cis would elute identically to an 18:1w7cis, which is itself a biomarker of gram- bacteria. Mono-unsaturated compounds with differing unsaturation positions could not be positively distinguished on a GC-IRMS trace alone; even distinguishing these on a GC-MS would be difficult due to similarity in chromatography. Further separation steps could have been undertaken (e.g., use of silver ion SPE) to separate mono- and poly-unsaturated FAMEs from each other, but even this method can be tricky implement.

b. Putting aside the difficulty in distinguishing FAME isomers based on retention time, many of the compounds identified in this study are absent from the BAME-CP and 37FAME standard mixtures that were used for retention time comparison, so I am confused how these compound identifications were made. For example, how were the 10-Me branched FAMEs identified? Or the 16:1w7 and 16:1w5? It seems that a large portion of this study hinges on accurate attribution of FAME production to specific taxonomic groups, and so making sure these attributions are correct is crucial to the outcomes of the study.

c. Another issue is the fuzzy relationship between source organism and FAME. As the authors state, only broad-level taxonomic identifications are possible based on fatty acid identity, and so they only identify compounds belonging to broad categories of microbes. However, the authors unilaterally ascribe specific biomarkers to fungi, bacteria, etc., without regard for the fact that many of these FAMEs are produced by multiple groups. For example, the authors cite 18:2 and 18:1w9 as a fungal biomarker, but an array of bacteria are known to produce these compounds. As stated previously, unsaturated FAMEs with different unsaturation positions would not be distinguishable on a GC-IRMS trace, and therefore would be falsely assigned to specific taxonomic groups. The severe risk for the confusion of biomarkers from different microbial groups calls into question the main finding of the study, as bacterial/fungal contributions (and some of those of bacterial subgroups such as actinos/gram+/gram-) cannot be disentangled. Substantial GC-MS work is needed to show that the peaks measured on IRMS are pure mixtures of a specific biomolecule – and even then, minor contributions from a highly-enriched non-target analyte could significantly alter the 2H composition of any given IRMS peak (e.g., a highly enriched 18:1w6 co-eluting with a less enriched 18:1w9).

d. The authors could consider identifying and measuring ergosterol, an unambiguous fungal biomarker, but this would not be a trivial undertaking and is outside the scope of this paper submission.

5. I do not believe the quantification of lipid abundance reported in this study is valid or in-line with standard practices in organic geochemistry/lipidomics. Line 562: The authors quantify their PLFAs and NLFAs relative to internal standard FAME 19:0, which was added just before derivatization to FAMEs. If this is the only internal standard added, it neglects the two major sources of uncertainty in lipid quantification: organic extraction efficiency and derivatization efficiency. An internal standard should be added prior to organic extraction in order to estimate the effectiveness of the extraction protocol (i.e., what percentage of PLFA/NLFA in the soil can be recovered by the method). Additionally, a separate internal standard should be added prior to derivatization in order to estimate the efficiency of the transesterification reaction (i.e., what percentage of PLFA/NLFA are successfully derivatized). It appears that neither was added. Furthermore, because 19:0 was already a FAME before the transesterification reaction, it cannot be used as a derivatization efficiency standard. This is a

crucial oversight, because the calculations of growth rate rely on accurate quantification of PLFA/NLFA abundance: the “Total PLFA C” term in Equation 2.

a. More nitpicky, but a specific quantification standard (separate from extraction and derivatization standards) would be useful to run, added immediately before GC analysis, rather than before derivatization.

Section-Specific Comments

Results

Line 177: Cell division rates are mentioned but never calculated in this study. It would be useful to readers to calculate inferred growth rates (days^{-1}) or doubling/generation times (days), which should be possible with your dataset with both 2H-PLFA and 18O-DNA data. This would help readers by allowing comparison of your data to past studies.

Lines 257 – 271: Is it known whether fungal storage components (triglycerides) use the same FAs as fungal membrane biomarkers (PLFAs)? If so, how do we know that NLFAs measured in this study are not just hydrolyzed PLFAs? If not, what NLFAs were detected and how do they differ from PLFAs?

Discussion

Lines 314 – 320: I am unsure what you mean by “PLFA composition alone (based on PLFA abundance) does not reliably describe the active fraction of the microbial community.” Do you mean that (1) the PLFA profiles do not match with what one would expect from sequencing, or (2) that greater PLFA abundance does not necessarily indicate greater growth rates? If (1), how are you determining the active fraction of the community without sequencing the DNA recovered? An approach similar to Koch et al. 2018 <https://doi.org/10.1002/ecs2.2090> would be required. If (2), a recent paper noting the decoupling of lipid-production and abundance is missing from this section (reference 35 (duplicate reference 80)) and should be discussed here as your conclusions are in agreement. In either case, a plot comparing PLFA/NLFA abundance with isotopic enrichment would be useful.

Lines 316 – 317: “While PLFAs can have a fast turnover time in soil, our results indicate that this might not hold true during drought conditions.” I would believe that PLFAs degrade more slowly in drought conditions, but I am unsure how your data proves this. You only measure PLFA production; how do you infer that PLFA degradation is slowed during drought? I don’t think this can be established by production rates alone.

Lines 394 – 396: It seems odd to me that a water assimilation factor is applied for lipids but not DNA. Is this the case? Are there estimates of water assimilation factors for DNA? I would believe that varying the a_w value for lipids affects the growth calculations, but wouldn’t this be true for DNA as well? Applying a factor (even an estimate) could affect your conclusions regarding the similarities/differences of growth rates measured between these methods.

Methods

Lines 542 – 545: See points above regarding attributing compounds to certain taxa. Review by Willers et al. 2015 <https://doi.org/10.1111/jam.12902> may be useful here (and useful to cite).

Lines 504 – 628: Total microbial biomass is measured via chloroform fumigation extraction (CFE), a standard method for determining “total microbial biomass carbon” in a soil. However, recent work (Mason-Jones et al. reference 30) suggests that CFE may be an underestimate because it overlooks storage compounds including triglycerides. So, it is unclear whether the calculations of Growth_NLFA (line 569) are accurate. Multiplying $(\text{NLFA C}_{\text{produced}})/(\text{Total NLFA C}) * \text{Microbial BMC} / \text{time}$, where Microbial BMC does not include triglycerides, seems risky, but I may be missing something. At least, this feels like a caveat that should be explicitly stated.

Throughout: As the authors say, the extent of isotopic labeling is determined relative to the isotopic enrichment of the 18O and 2H tracer solutions. They accomplish this by generating equilibration curves (Supplementary Text/ Fig. S8) and integrating the isotopic enrichment over 48 hours, taking the average value as the effective isotopic tracer strength. Similar to point #1, it is necessary to state upfront the values of the effective label strength and how they varied across the soil conditions. It is also necessary to show raw 2H/18O enrichment of PLFA/NLFA over time so that readers can assess where the greatest rate of enrichment occurred.

Throughout: I am also curious the extent to which vapor equilibration homogeneously labels the soil. I imagine water would penetrate from outside to inside of soil aggregates and therefore soil microbes existing on the periphery of an aggregate would be labeled more rapidly than those in interior spaces. Could you comment on whether this can be assessed and the degree to which it could play a role in your measurements?

Line Edits

Line 40 – abstract: “Community-level CUE (the balance between anabolism and catabolism) was unaffected by drought but decreased in future climate conditions.” The wording in this line is unclear. One would expect that drought is a possible future climate condition for many biomes. I know this is further discussed, but for the abstract, it would be good to clarify what the future climate condition entails that decreased CUE. In addition, a CUE decrease may not be intuitive to readers unfamiliar with the term – does a decrease favor anabolism or catabolism? It would be helpful to specify that here.

Line 122 – abstract: “all microorganisms synthesize lipids and fatty acids to build and maintain cell membranes regardless of

their metabolic activity and cell cycle stage” is a bit misleading. While 2H-lipid-SIP quantifies anabolic activity by tracking the production of membrane lipids, this production rate does likely relate to metabolic activity and the metabolic production of free energy on some level. Perhaps it is more accurate to say, “all microorganisms synthesize lipids and fatty acids to build and maintain cell membranes regardless of their metabolic strategy and cell cycle stage.”

Supplementary Figure S10: In my view, the more appropriate comparison here is not a condition-specific paired t-test, but rather a comparison of the 1:1 line and a linear regression of the data plotted in panel B. To my eye it looks like there could be an offset between 1:1 line and a line of best fit through the data. I am not very concerned with deuterium toxicity at the enrichments applied in this study, but this would be useful to plot in order to prove it to readers, especially because your deuterium enrichments are relatively high.

Suggestions for reframing

The major concerns I have outlined above, in my opinion, should preclude this work from publication in its current form. However, I do not want to imply that I see no value in the experiments conducted here. The authors could benefit from stepping back from taxonomically-resolved estimates of growth (e.g., fungi vs. bacteria, gram+ from gram-, etc.) and focusing on the community level measurements. How do the community-level growth rates compare to other lipid-SIP and DNA-SIP estimates of growth (see meta-analysis by Koch et al. 2018 for example). How do the values derived from the vapor-SIP method compare to studies where the soils are rewetted or slurried? How does the drought condition compare to previous studies of soil rewetting? These are all interesting scientific questions that can be answered with the dataset in hand and would be valuable for the community. Such a manuscript would be appropriate for a targeted soil microbio or environmental microbio journal.

The comparison of community-level production of PLFAs and DNA, inferred from 2H and 18O, also provides opportunities to discuss soil microbial physiology, specifically how growth is manifest in this system and balanced between nucleic acids and membrane lipids. More discussion here is warranted regarding how these rates relate to each other and to respiration. Indeed, it does seem that there exists substantial offset between PLFA and DNA growth rates. This is intriguing! Could this difference point to differences in biomolecular investment? Or does this speak to the nature of the methods: DNA is replicated in a more binary fashion than lipids are produced (more continuously). Focusing on these aspects would pull the scope of the paper away from the more fraught/risky aspects of taxon-specific growth rate assignment and more towards a comparison of DNA and lipids as growth markers at the community-level.

Reviewer #3

(Remarks to the Author)

This paper by Canarini et al. investigates the response of soil microorganisms to future climate scenarios, including increased atmospheric CO₂ concentration, increased temperature, and drought. Using a novel isotope-based method, the authors can partition microbial growth rates into bacterial and fungal growth rates and explore their different responses to environmental change. The authors also explore the carbon allocation of microbial communities into storage compounds in the different climate change scenarios.

The authors use an innovative method that allows them to answer an important question: how do fungi resist long periods of drought? Lots of literature suggests that fungi are more resistant than bacteria to drought, however not many studies show the mechanisms regulating this, which Canarini et al. do.

Overall, I found that the manuscript was well written, clear, and easy to follow. The figures and legends are clear, and so is the text.

I have some main concerns/questions about the study:

1) Authors express all their growth and respiration rates as mass specific rates, and I am unsure why is that. I believe that if the contribution of microbes to C fluxes wants to be understood the rate should be an absolute rate. Treatments might affect biomass and functions differently and thus it is difficult to disentangle effects and understand how the environment is driving microbial mediated C cycling. So, the changes in mass specific rates are they due to changes in rates or changes in biomass? Or both? I suggest at least indicating mass specific rates as well as rates per g of dry soil, followed by a discussion of the differences observed.

2) The authors incubate samples at field temperature. How can we know that the results they see is not an effect of the differences in temperature? Have the authors account for that effect?

3) I would like to see the continuous data of the field experiment rather than the end point moisture level (Table S6), to understand the moisture history the soil microorganisms have been exposed to.

In line comments:

Line 83. CUE is defined as C allocated to growth relative to respiratory processes. Perhaps, C allocated to growth relative to the total amount of C used by microorganisms would be more appropriate.

Line 85. I understand why authors here refer to growth. However, there is not a clear link back to CUE. If the aim is to reach a broader audience that might not be familiar with this this might be a bit confusing for the reader.

Lines 93-94. What happens with growth when temperatures increase?

Line 93. In which direction? Please provide examples.

Lines 101-102. Soil functions and microbial communities. Which functions and aspects of the microbial communities are the authors referring to? If the aim here is to point out a research gap I advise to be more specific.

Figure 1 and in general. Why are the medians indicated instead of the average?

Line 281-282. Please add reference.

Line 300-301. How severe/intense is the drought for the microbial communities in this ecosystem?

Line 364-365. This could be due to the time frame where the recovery was measured. In addition, did authors measure growth and respiration immediately after sampling? That could also have an effect. Please clarify how much time passed between sampling and measuring.

Line 404. "During" has been written twice.

Lines 445-446. Please clarify how long between sampling date and measuring dates (see above).

Lines 475-476. Why did authors decide to incubate at different temperatures instead of a single common temperature? How can authors disentangle the direct temperature effect from the field treatment effect? (See above)

Lines 510-512. This is not clear, which specific growth rates have to be considered? Please clarify.

Line 545. It is not clear whether authors use actinobacteria as Gram + bacteria when looking at fungal to bacterial ratios.

Version 1:

Reviewer comments:

Reviewer #1

(Remarks to the Author)

The authors have done a great job revising their manuscript. The manuscript will make a nice contribution to the literature. Congratulations!

Reviewer #2

(Remarks to the Author)

Canarini, Fuchslueger, et al.,

The authors have made substantial progress on this manuscript. The authors have elegantly addressed the majority of my concerns, and I commend them for creating an eloquent and important piece of literature.

Regarding my suggested reframing of the manuscript, the authors have sufficiently defended their current framing and provided much needed methodological context and caveats to their main text. I can accept the current framing of the revised paper and thank the authors for their diligence in addressing my comments.

I believe the majority of my comments have been addressed, including all of my minor and line comments. However, I must continue to raise concerns regarding my major comments No. 1 and No. 5. I do not think my remaining concerns preclude this work from publication, but I would like to see them addressed, if possible, before this work enters print.

Major Comments:

1. Inclusion of labeling data.

"We now include further information both on natural abundance and $2\text{H}/18\text{O}$ tracer incorporation in the supplementary materials ..."

Thank you for the inclusion of the average isotopic composition data. This will be useful for readers in contextualizing the extent of labeling. While I think this data is useful to include in the main text, it doesn't address the point at-large. Readers and those designing similar experiments, I think, need to know how specific compounds (individual lipids measured, DNA) became enriched over time (even if that is two timepoints, start and end). An average value, with no corresponding figure, does not convey heterogeneity in compound-specific labeling and may be obscuring highly labeled compounds or poorly labeled compounds. I see the authors have included their compound-specific labeling data as a supplementary dataset, but I would suggest a figure or SI figure, so that this data is more central to the story.

2. Free fatty acids in the environment.

"The applied derivatization method does not derivatize free fatty acids and therefore those are excluded from the analysis" Thank you for clarifying this in the main text and providing citation. The authors have addressed this concern.

3. Slower triglyceride degradation.

"We agree with the reviewer that we would underestimate mass-specific NLFA production rates due to possible necromass-NLFA. However, in Fig.4b we present results of total NLFA produced (total amount) as percentage of PLFA (also total amount)..."

The authors have addressed this concern by explicitly stating this caveat in the text. I would also suggest including the caveat in the figure caption (Fig. 4) as an explanation to why the NLFA/PLFA ratio is plotted in panel B. I suppose the community needs to start measuring the degradation rates of different lipid classes in soil!

4. Compound identification via GC

Thank you for addressing my concern regarding the standards mixtures. I hope the authors can appreciate the alarm of seeing identified compounds that were not present in the standards. The authors have included sufficient information,

including that they used GC-TOF-MS to further identify compounds. Seeing the example spectra has helped the authors prove their point, assuming this spectrum is a soil sample and not a library of mixed standards (by eye it looks like a typical soil profile...).

By addressing my concerns on compound ID via GC, the authors have also assuaged my concern of taxonomic ID. In brief, if the GC peak identification was spotty, then I would have serious concerns about their taxonomic assignment. However, because it looks like the authors have done their due diligence in building a FAME compound library and conducting MS identification, I can believe their taxonomic assignment follows the best knowledge available in the literature.

5. Quantification of lipid abundance.

I remain concerned about the authors' calculation of PLFA growth. I understand how the authors are calculating relative PLFA production without extraction and derivatization standards, as well as how the Lipid_{produced} and Lipid_{total} terms cancel out in their calculations and are multiplied by the total microbial biomass carbon (MBC) as estimated by chloroform fumigation extraction (CFE). However, the assumption the authors are making here is that lipid and biomass C (measured by CFE) is a 1-to-1 relationship. Because the authors did not include extraction standards, they cannot know with confidence how much PLFA was present in their samples per gram of soil. They only know the total MBC from the CFE. Of course, there is an intuitive link between lipid content and total MBC, but these do not exist at a one-to-one relationship and there may be a substantial offset between the two (e.g., see Bailey et al. 2002 [https://doi.org/10.1016/S0038-0717\(02\)00070-6](https://doi.org/10.1016/S0038-0717(02)00070-6)). Therefore, production rates of PLFA may not adequately translate to MBC. To avoid confusion, I would advise the authors stick to relative growth rates, not normalized to mass, (e.g., day⁻¹, yr⁻¹, etc.) as their primary reported metric. The authors have stated that their intention is to focus on relative differences between the treatments, which I think is more readily accomplished with intuitive units of measurement that are already widely reported in the literature.

Minor comments:

Please check the rendering of your equations, some exhibit blank spaces.

Version 2:

Reviewer comments:

Reviewer #2

(Remarks to the Author)

I thank the authors for their diligence in responding to the reviewers' comments and concerns. I look forward to seeing this work in print.

REVIEWER COMMENTS

We thank the three reviewers for their thorough reviews and the insightful suggestions, and also want to thank the editor for providing a chance to improve and strengthen our manuscript.

Our point by point answers are given in blue and the line numbers refer to the new manuscript version without track changes. For clarity we also included a version with track changes.

Reviewer #1 (Remarks to the Author):

The paper by Canarini et al. uses D2O labelling to investigate the synthesis of membrane and storage lipids of soil microbes. They made use of a field experiment with “future climate” and drought treatments. Hence the design allowed them to test how microbial ecophysiology is affected by drought and future climate. The main novelty of the work is application of D2O labeling and subsequent measurement of 2H incorporation into membrane and storage fatty acids. Critically, they used a vapour phase approach to label soil and thus, unlike liquid D2O addition, they managed to estimate synthesis without affecting soil water content. Their work revealed that fungi remained active and invested into storage compounds under the drought conditions. The experimental work is thorough and data interpretation is fine, so most of my comments relate to use of literature and tightening concepts.

One of the main findings is that fungi are less affected by water deficits than bacteria, but interpretation would benefit from considering the severity of water deficit. I recommend bringing water deficit to the fore to help contextualise the differential responses of bacteria and fungi. Was the stress severe or mild? What sort of water potentials were involved?

This is an excellent point. The drought treatment decreased the soil water content continuously from mid June until the rewetting at the end of July. At the time of sample collection the soil water content ($\text{g H}_2\text{O g}^{-1}$ dry soil) was strongly decreased with values below 0.1 compared to values around 0.3 in the non-drought treated ambient and future-climate plots. We added information on the drought intensity upfront in the result section (L. 183-187). As suggested by reviewer n.3, we have also added information on continuously measured volumetric soil water content data for all treatments over the course of the experiment in the supplementary section (now Fig. S1, see also below, and related method description in L. 480-484) and rearranged all the supplementary figures accordingly. Moreover, in the discussion we emphasized again the relative decrease in soil water content in the drought compared to the respective controls in order to contextualize the extent of the drought treatment (now L. 318).

Fig. S1. Soil water content. Daily averages of volumetric soil moisture (%) at a) 3 and b) 9 cm of soil over the course of the experiment (June-August 2020). The line colours indicate the different treatments.

In quite a few places it seemed PLFA was being used inappropriately. The first rather minor issue is that PLFA is a misnomer given that the fraction can contain fatty acids from lipid classes in addition to phospholipids. I don't think here is the place to introduce new terminology because PLFA is firmly entrenched in the literature. Nevertheless, in the interest of accuracy, avoid phospholipid and instead speak of membrane lipid.

We agree with the reviewer and are aware that with methodology used the collected PLFA fraction may encompass other membrane lipid classes. Previous studies have found that these other classes of lipids (mainly betaine lipids) should represent a low percentage of the total amount in non-P deficient soils (around 3-6%; Warren, 2019). In addition, not all membrane lipids are present in the PLFA fraction, as for example Glycolipids, after fractionation, are separated from PLFA. For consistency with previous literature we decided to keep the PLFA terminology, but, following the reviewer's suggestion, we added a clarification to make readers aware that other classes of lipids might be present with the PLFA fraction.

L.500-502, "Although the extracted PLFA fraction may contain other lipid classes, representing minor components of membrane lipids, we use the term PLFA throughout the manuscript as this is the major fraction analysed".

Warren CR. 2019. Does silica solid-phase extraction of soil lipids isolate a pure phospholipid fraction? *Soil Biology and Biochemistry* **128**: 175–178.

The more confusing second issue is that “PLFA” is in some places used to refer to fatty acids from storage lipids. This invites confusion and could mislead into thinking fatty acids of membrane and storage lipids were measured simultaneously.

We have rephrased and modified several sentences to be more specific:

In L. 135, we added: “*which are separated from PLFA after extraction from soil*”

In L. 385 we changed “²H-PLFA-SIP” to “²H-labelling”.

Line 317-. Speaks of turnover of membrane lipids in soil, but seems to misinterpret the data. Available data show that turnover of phospholipids is very rapid after death or when added to soil (reference 54). However, the turnover of membrane lipids within microbes is far, far slower (ref 78). Likely due to C-H of lipids (and presumably much of the fatty acid backbone) being re-used multiple times.

We agree with the reviewer and understand that further data would be necessary to prove this. Following reviewer 2’s comment we removed the statement from the manuscript.

It’s a moot point if the comparison of DNA vs fatty acid production adds a lot to the manuscript. It’s definitely an interesting methodological question, but additional data are required to tease apart the reasons for differences between methods. I say this because there are many unknowns and assumptions in both approaches. Without additional mechanistic insights the conclusion is a bit prosaic: the methods differ but we don’t know why or which is more accurate.

We agree with the reviewer that explaining the differences in absolute growth rates obtained by the two methods is not possible with the available data. Our intention was rather to demonstrate that the experimental treatments (future climate and drought) have similar impacts on microbial activity (as shown by the statistical analysis and average differences), using two independent methods. In response to the reviewer we restructured some parts of the results section (see line 187-206). We think that both methods provide very interesting insights into responses of soil microbial communities to climate change, but we tried to omit any discussion on ranking the methods. For clarity, we noted that the absolute values differ (L. 418-419) and try to explain some reasons for such differences. Nevertheless, we primarily aimed to highlight the high consistency between the two methods when comparing treatment effects (e.g., now L. 414-416).

Ultimately the largest challenge is that recycling of C-H bonds of lipids means growth is not wholly dependent on lipid synthesis. Also, the discussion of factors affecting ²H enrichment may be conflating (or combining?) two different factors. First there is the fractionation that can occur between source water and C-H in a lipid. The independent second factor is the fraction of C-H bonds in a lipid that can be labelled.

This is indeed an important point. We tried to rephrase the discussion on the factors affecting ²H incorporation to address the reviewer’s comment. We highlight that both factors that the reviewer mentions may play a role: fractionation of H during assimilation, and the fraction of H that can be labelled via water isotopes. Because these two factors cannot be disentangled with our measurements, these are accounted for via an assimilation factor (a_w). We hope that the re-phrased sentence explains this more clearly. See below:

L. 414-417: *“However the amount of ^2H derived from added labelled water can be considered as a combination of two factors: the mole fraction of water derived H that can be incorporated into fatty acids and the associated net ^2H isotope fraction.”*

Minor comments

Line 123. There are quite a few other studies that have used D2O labeling to investigate synthesis of lipids in soil:

- Huguet, A., Meador, T.B., Laggoun-Défarge, F., Könneke, M., Wu, W., Derenne, S., Hinrichs, K.-U., 2017. Production rates of bacterial tetraether lipids and fatty acids in peatland under varying oxygen concentrations. *Geochimica et Cosmochimica Acta* 203, 103-116.
- Warren, C.R., 2023. LC-MS analysis of D2O-labelled soil suggests a large fraction of membrane lipid exists within slow growing microbes. *Soil Biology and Biochemistry* 177, 108912.
- Wegener, G., Bausch, M., Holler, T., Thang, N.M., Prieto Mollar, X., Kellermann, M.Y., Hinrichs, K.U., Boetius, A., 2012. Assessing sub-seafloor microbial activity by combined stable isotope probing with deuterated water and ^{13}C -bicarbonate. *Environ Microbiol* 14, 1517-1527.
- Wegener, G., Kellermann, M.Y., Elvert, M., 2016. Tracking activity and function of microorganisms by stable isotope probing of membrane lipids. *Current Opinion in Biotechnology* 41, 43-52.

We thank the reviewer for the suggestions. We added all the mentioned citations.

Line 36. Rather than “fatty acids”, I think there’s a need to specify membrane and storage lipids.

We have changed this accordingly.

Line 126. Not correct. PLFA-SIP can measure only membrane lipids. A separate (not concurrent and not PLFA) measurement is required for measuring production of storage compounds. Also, reference 38 does not contain data that support this point.

We thank the reviewer for spotting this point. We rephrased (L.130) and added a sentence and the relevant reference to make this clear (now L. 136).

L. 130: *“The major advantage of tracing ^2H into fatty acids (^2H -FAME-SIP) is that it allows the simultaneous and sensitive quantification of bacterial and fungal replication rates.”*

L. 136: added citation nr. 40:

*“Gorka, S. et al. Beyond PLFA: Concurrent extraction of neutral and glycolipid fatty acids provides new insights into soil microbial communities. *Soil Biology and Biochemistry* 187, 109205 (2023).”*

Furthermore, to fully address the reviewer’s comments and reflect the fact that the method encompass both PLFA and NLFA, we modified the name to ^2H -vapor-FAME-SIP (FAME as in “fatty acid methyl ester”) to more broadly refer to a methodology enabling to measure production rates by targeting ^2H in fatty acids.

Line 134-135. Reference 38 does not contain empirical data so doesn’t really support this point.

We are sorry for this mistake. We replaced the reference with: “Mason-Jones, K., Breidenbach, A., Dyckmans, J., Banfield, C. C. & Dippold, M. A. Intracellular carbon storage by microorganisms is an overlooked pathway of biomass growth. *Nat Commun* **14**, 2240 (2023).”

Line 154, hypothesis 3. I think the logic needs explaining. Is the logic that water deficit reduces nutrient supply more than C and the increase in C:nutrient leads to storage? Or is the logic related to dormancy?

We added a sentence in the introduction to make this clearer:

L. 146-150 : “During drought conditions dissolved organic C has been shown to accumulate in soil pores, increasing its concentration in the remaining water. At the same time decreased moisture reduces the connection and mobility of substrate to microbes, which might lead to higher C investment into storage compounds.”

We hope that this provides the readers with sufficient information to understand the rationale behind the formulation of our hypothesis.

Fig 2 legend. It’s not clear to me what data were used for the PCA. E.g. Is it rate of 2H enrichment, or rate of synthesis?

We used mass-specific rates of synthesis. We now clarified in the legend of Fig. 2

Reviewer #2 (Remarks to the Author):

The work by Canarini et al. builds upon recent advances in ^{18}O -DNA stable isotope probing and deuterium-based lipid stable isotope probing to measure the biomass production rates of fungal and bacterial lipid compounds in soil. The authors provide a valuable comparison between these methods to measure incorporation of ^2H and ^{18}O into lipid and DNA pools, respectively, which can relate the production rates of lipid and nucleic acid biomass, and their relationship to respiration rate.

Furthermore, the authors examine the neutral lipid fraction to measure the production rates of storage compounds – namely FAs derived from triglycerides. All of these measurements are undertaken in the context of future climate conditions including elevated CO_2 ($e\text{CO}_2$) and drought. The authors include monitoring of microbial respiration processes in order to estimate carbon use efficiency (CUE) in their soil conditions. The positioning of this study in the scientific literature is important and the results of such a study would be of interest to soil microbiology community. I find the comparison of lipid-SIP and DNA-SIP to be very timely given heightened research in both of these areas within the past few years. The use of vapor-based tracers represents a further advancement towards more passive measurement of microbial activity in situ. The authors infer logical conclusions from their data regarding the effects of drought and future climate conditions on microbial growth and storage and I have no issues with their discussion or how it informs soil C dynamics more broadly.

Unfortunately, while the conclusions drawn on the data are valid, I have serious concerns regarding the data itself and the methodology, which does not meet the expected standards of the organic geochemistry field. In the paper's current form, major sources of uncertainty remain unaddressed and, if unanswered, would leave the conclusions of the study unjustified. I cannot recommend this work for publication at Nature Comms without the authors undertaking substantial work addressing the following issues, which, at the discretion of the editor, could be outside the scope of the current submission.

My intention with writing this review is not to be overly harsh to the authors. Indeed, I was excited to see the preprint of this work as it represents the first study (to my knowledge) to co-register DNA- and lipid- SIP strategies, and certainly the first to do it with vapor equilibration. I identify deficiencies mainly in relation to the core lipid-SIP measurements of the study because these problems are glaring and would be quickly noted by microbiologists familiar with soil organic geochemistry. At the end of my review, I suggest potential ways the authors can reframe and improve the manuscript.

We thank the reviewer for both the kind words regarding the importance of our study, as well as the critical comments on the methodology and possible consequences on the interpretation of results. We have now added more details on our methodology and believe that we have addressed all the reviewer's comments. .

In the following we further address each comment point by point, provide supporting data as requested and add more details on the methodologies. We look forward to the evaluation of our comments and thank the reviewer for such a thorough revision and valuable suggestions.

Major Comments

1. I am concerned that there is no reference to or description of the isotopic values of the PLFAs, NLFAs, and DNA held within the main text, supplementary texts, or SI data. This is

critical context for readers to contextualize the extent of isotopic labelling that has occurred. I would recommend a figure, either in main text or supplement that shows, for each incubation condition, differentiating specific compounds (specific PLFAs, NLFAs, and DNA), the observed increase in ^2H and/or ^{18}O over time. Including raw data as a supplementary table would also be useful for reproducibility. The natural abundance isotopic values of these compounds in your study site would need to be included, as well as the effective isotopic label strength the samples experienced. This is important to include for multiple reasons. First, quantitative isotopic labeling occurs only when i) sufficient isotopic tracer is incorporated above background noise and ii) the extent of incorporation does not approach that of the label. Without this information, readers cannot assess the precision of your quantification. Second, because D_2O enrichments of > 20 at. % were applied to the soils, it is possible that extremely high ^2H enrichments of FAMES were achieved. If this were the case, I would have further concerns regarding the quality of IRMS data (e.g., mass 3 and mass 2 peaks beginning to overlap, risk for chromatography shifts due to multiply-substituted FAMES, memory effects, etc.), but without the data presented, neither I nor future readers can assess these factors. I consider this a major omission that must be addressed prior to publication.

We now include further information both on natural abundance and $^2\text{H}/^{18}\text{O}$ tracer incorporation in the supplementary materials, as well as the amount of label that the samples were subjected to, over the course of the incubation (L. 541-549).

These data confirms that: i) we have achieved sufficient isotope tracer incorporation in both PLFA and DNA to exceed background noise and ii) it did not reach the maximum tracer applied. While we had a relatively high ^2H incorporation, we have valid chromatographic resolution (see example chromatograms from a ^2H labelled and a natural abundance sample below). For ^2H we obtained an average biomarker isotopic value across labelled samples of 0.207 at% for ^2H in PLFA (natural abundance value averaged at 0.0136) and of 0.0609 at% for ^2H in NLFA (natural abundance value averaged at 0.014). For ^{18}O in DNA we obtained an average isotopic value of 0.214 at% (natural abundance value averaged at 0.203). We now added this to the manuscript in L. 604-606 and in L. 671-673.

2. Regarding NLFAs, it is unclear to me how the authors differentiate between triglyceride-sourced NLFAs (from storage compounds) and free fatty acids in the environment. Free fatty acids, sourced from both living and dead biomass from all domains of life, would certainly be present in these soils. During organic extraction and separation, free FAs would co-elute with FAs sourced from triglycerides in the NLFA fraction. In the methods section, no mention is made whether free FAs were quantified and distinguished from triglyceride-FAs. The conclusion that is most at risk of this problem is the finding that fungi produce NLFAs during drought. Free FAs in the environment can be sourced from microbial biomass as well as eukaryotic biomass (e.g., degraded plant tissue) and these free FA concentrations could vary widely between soil types and conditions. It also seems possible to me that PLFAs could have been produced (by fungi or others), these cells lyse, causing their PLFAs to hydrolyze to free FAs, and these free FAs eluting in the NLFA fraction would be mistakenly attributed to triglycerides. This risk may be high due to the rapid degradation of PLFAs (on the order of hours – see cited Zhang et al.).

The applied derivatization method does not derivatize free fatty acids and therefore those are excluded from the analysis (see for example citation below). We now added the following explanation:

L. 575-576: “This method does not derivatize free fatty acids, which are therefore not included in the analysis”.

Chowdhury, T. R. & Dick, R. P. Standardizing methylation method during phospholipid fatty acid analysis to profile soil microbial communities. *Journal of Microbiological Methods* **88**, 285–291 (2012).

3. Another point regarding NLFAs: the major assumption inherent to all PLFA studies is that PLFAs degrade rapidly upon cell death (the authors appropriately cite the recent Zhang et al. work on this). PLFA degradation enables their use as conservative tracers of microbial biomass production because it targets analyses at living biomass. I am unaware of any study measuring the triglyceride degradation rate in soil. (From a quick literature search on my end, it appears that these rates in soil are unconstrained). Due to a lack of a polar head group, one would expect that rates of triglyceride degradation would be far slower than those of PLFAs. Therefore, it seems fraught to present NLFA/triglyceride production rates without commenting on the strong possibility that a high concentration of NLFAs are sourced from necromass. If the measured NLFA pool contains substantial necromass, calculated rates of NLFA-related growth would not be quantitative, as they would be dependent on the amount of relict NLFA necromass in each soil condition. At the minimum, the authors would need to state upfront the caveat that their NLFA production rates are under-estimates due to the relict FAs present in the environment and the fact that triglycerides from dead cells are not rapidly lost from the NLFA pool.

We agree with the reviewer that we would underestimate mass-specific NLFA production rates due to possible necromass-NLFA. However, in Fig.4b we present results of total NLFA produced (total amount) as percentage of PLFA (also total amount), and therefore is not affected by the potential NLFA higher necromass. We believe this offers a different perspective on this process and overcomes the potential limitation on quantitative estimates of production rates raised by the reviewer and allows a better comparison across treatments. We thank the reviewer for pointing out that this can be misunderstood, and added more discussion on the potential slower degradation of NLFAs to fully address the reviewer's comment (L. 380-385).

4. I have major concerns regarding compound identification and ascription to microbial groups:

I am concerned that the authors' compound identifications may be incorrect, which would call into question their ability to distinguish microbial taxonomic groups. The authors state on lines 542 – 545: "We used the markers 18:1w9cis and 18:2w6,9 to estimate fungal biomass" and continue to state the markers used for gram+ and gram- bacteria. The authors identify these compounds using the retention times of BAME-CP and 37FAME standard mixtures (line 530). The issue here is that a GC-IRMS does not provide structural information regarding the unsaturation of the FAME compounds. For example, an 18:1w9cis would elute identically to an 18:1w7cis, which is itself a biomarker of gram- bacteria. Mono-unsaturated compounds with differing unsaturation positions could not be positively distinguished on a GC-IRMS trace alone; even distinguishing these on a GC-MS would be difficult due to similarity in chromatography. Further separation steps could have been undertaken (e.g., use of silver ion SPE) to separate mono- and poly-unsaturated FAMES from each other, but even this method can be tricky implement.

Putting aside the difficulty in distinguishing FAME isomers based on retention time, many of the compounds identified in this study are absent from the BAME-CP and 37FAME standard mixtures that were used for retention time comparison, so I am confused how these compound identifications were made. For example, how were the 10-Me branched FAMES identified? Or the 16:1w7 and 16:1w5? It seems that a large portion of this study hinges on accurate attribution of FAME production to specific taxonomic groups, and so making sure these attributions are correct is crucial to the outcomes of the study.

Another issue is the fuzzy relationship between source organism and FAME. As the authors state, only broad-level taxonomic identifications are possible based on fatty acid identity, and so they only identify compounds belonging to broad categories of microbes. However, the authors unilaterally ascribe specific biomarkers to fungi, bacteria, etc., without regard for the fact that many of these FAMEs are produced by multiple groups. For example, the authors cite 18:2 and 18:1w9 as a fungal biomarker, but an array of bacteria are known to produce these compounds. As stated previously, unsaturated FAMEs with different unsaturation positions would not be distinguishable on a GC-IRMS trace, and therefore would be falsely assigned to specific taxonomic groups. The severe risk for the confusion of biomarkers from different microbial groups calls into question the main finding of the study, as bacterial/fungal contributions (and some of those of bacterial subgroups such as actinos/gram+/gram-) cannot be disentangled. Substantial GC-MS work is needed to show that the peaks measured on IRMS are pure mixtures of a specific biomolecule – and even then, minor contributions from a highly-enriched non-target analyte could significantly alter the 2H composition of any given IRMS peak (e.g., a highly enriched 18:1w6 co-eluting with a less enriched 18:1w9).

The authors could consider identifying and measuring ergosterol, an unambiguous fungal biomarker, but this would not be a trivial undertaking and is outside the scope of this paper submission.

We thank the reviewer for pointing out the carelessness of not fully reporting the methods we used. PLFA/NLFA extraction and analysis is routinely done in our laboratory for decades and we have a quite comprehensive library on FAME compounds extracted from environmental samples (e.g., Fuchslueger *et al.*, 2014; Kaiser *et al.*, 2015), as well as an array of individual organisms from pure cultures (e.g., as published in Gorka *et al.*, 2023), including arbuscular mycorrhizae, ectomycorrhizal, saprotrophic fungi as well as bacterial cultures of gram positive, negative and actinobacteria. We routinely run samples additionally on a GC-TOF-MS together with FAME and BAME standards using the same program and column as specified for the GC-IRMS. This allows us to correctly identify biomarkers from the chromatograms obtained from the GC-IRMS. If peaks still cannot not be identified, we do not assign them to a specific group.

We added this information in the method section (now L.583-592) and report here below an example chromatogram from the GC-TOF-MS (all peaks and zoom to the region with C18 and C19 peaks) as well as from the GC-IRMS (attached above to a previous comment).

Peak overview

Zoom

In our analysis we have achieved a sufficient separation of the biomarkers analysed by GC-IRMS (see figure shown earlier). To further address the reviewer's concern about a possible contamination of one of the fatty acids used to categorize fungi, we report the same results shown for the whole fungal group also for the two individual biomarkers used for fungi, to show that both indicate identical conclusions.

Finally, we would like to point out that it is well known that some biomarkers are expressed by several groups (see the same review suggested by the reviewer, Willers et al. 2015), but the

principle of the PLFA method is that fatty acids used to indicate groups are expressed in really high abundance in those groups. We believe we can show that our analysis can differentiate individual fatty acids, and that we conservatively assign fatty acids to microbial groups following the best knowledge available in the literature (Frostegård, Å. et al., 2011; Willers, C. et al., 2015; Quideau, S. A. et al. 2016; Gorka, S. et al. 2023).

Frostegård, Å., Tunlid, A. & Bååth, E. Use and misuse of PLFA measurements in soils. *Soil Biology & Biochemistry* **43**, 1621–1625 (2011).

Willers, C., Jansen van Rensburg, P. J. & Claassens, S. Phospholipid fatty acid profiling of microbial communities—a review of interpretations and recent applications. *Journal of Applied Microbiology* **119**, 1207–1218 (2015).

Quideau, S. A. et al. Extraction and Analysis of Microbial Phospholipid Fatty Acids in Soils. *JoVE* e54360 (2016) doi:doi:10.3791/54360.

Gorka, S. et al. Beyond PLFA: Concurrent extraction of neutral and glycolipid fatty acids provides new insights into soil microbial communities. *Soil Biology and Biochemistry* **187**, 109205 (2023).

Fuchslueger L, Bahn M, Fritz K, Hasibeder R, Richter A. 2014. Experimental drought reduces the transfer of recently fixed plant carbon to soil microbes and alters the bacterial community composition in a mountain meadow. *New Phytologist* 201: 916–927.

Kaiser C, Kilburn MR, Clode PL, Fuchslueger L, Koranda M, Cliff JB, Solaiman ZM, Murphy DV. 2015. Exploring the transfer of recent plant photosynthates to soil microbes: mycorrhizal pathway vs direct root exudation. *New Phytologist* 205: 1537–1551.

5. I do not believe the quantification of lipid abundance reported in this study is valid or in-line with standard practices in organic geochemistry/lipidomics. Line 562: The authors quantify their PLFAs and NLFAs relative to internal standard FAME 19:0, which was added just before derivatization to FAMES. If this is the only internal standard added, it neglects the two major sources of uncertainty in lipid quantification: organic extraction efficiency and derivatization efficiency. An internal standard should be added prior to organic extraction in order to estimate the effectiveness of the extraction protocol (i.e., what percentage of PLFA/NLFA in the soil can be recovered by the method). Additionally, a separate internal standard should be added prior to derivatization in order to estimate the efficiency of the transesterification reaction (i.e., what percentage of PLFA/NLFA are successfully derivatized). It appears that neither was added. Furthermore, because 19:0 was already a FAME before the transesterification reaction, it cannot be used as a derivatization efficiency standard. This is a crucial oversight, because the calculations of growth rate rely on accurate quantification of PLFA/NLFA abundance: the “Total PLFA C” term in Equation 2.

More nitpicky, but a specific quantification standard (separate from extraction and derivatization standards) would be useful to run, added immediately before GC analysis, rather than before derivatization.

We agree with the reviewer that our approach does not take extraction and derivatization efficiency into account. This is rarely done in soil lipid studies (see for example Rousk et al., 2013; Kallenbach et al., 2016; Qin et al., 2019; Butler et al., 2023). While we agree that this is an important topic, we argue that correcting for extraction or derivatization efficiency would not

influence the outcomes of growth rates of our experimental data, as we try to explain in more detail below:

First, extractions and derivatization were carried out on the same day for all the samples, and under exact same conditions. In more detail, the derivatization step is carried out in one 96 well plate (glass vials) where all the vials are subjected to the same exact conditions (1 well for PLFA and 1 well for NLFA inserted in the same water bath at the same time). This minimizes any between-sample differences, which are instead mostly derived from pipetting differences, for which our internal standard approach can accurately account for.

Second, in our study we report mass-specific microbial growth, which allows a more unbiased comparison of microbial physiological responses of differences among the applied treatments. Such mass-specific rates are independent of extraction or derivatization efficiency.

Third, as shown in equation 2, we multiply the ratio of new and total PLFA by the microbial biomass obtained by chloroform fumigation extraction, which eliminates any potential issue caused by not fully accounting for extraction/derivatization efficiency.

To fully address the reviewer comments and for clarity to the readers we also added these details in the method section, to make the readers aware that our approach does not fully account for extraction or derivatization efficiency:

L. 575-577 “The internal standard is added as FAME and therefore our approach does not fully account for extraction or derivatization efficiency. To minimize differences between samples, all the vials were derivatized in the same day using the same conditions.”

while it reliably measures mass-specific growth rates or community rates converted by the standard chloroform fumigation method.

REFERENCES:

Butler OM, Manzoni S, Warren CR. 2023. Community composition and physiological plasticity control microbial carbon storage across natural and experimental soil fertility gradients. *The ISME Journal*: 1–11.

Kallenbach CM, Frey SD, Grandy AS. 2016. Direct evidence for microbial-derived soil organic matter formation and its ecophysiological controls. *Nature Communications* 7: 13630.

Qin S, Chen L, Fang K, Zhang Q, Wang J, Liu F, Yu J, Yang Y. 2019. Temperature sensitivity of SOM decomposition governed by aggregate protection and microbial communities. *Science Advances* 5: eaau1218.

Rousk J, Smith AR, Jones DL. 2013. Investigating the long-term legacy of drought and warming on the soil microbial community across five European shrubland ecosystems. *Global Change Biology* 19: 3872–3884.

Section-Specific Comments

Results

Line 177: Cell division rates are mentioned but never calculated in this study. It would be useful to readers to calculate inferred growth rates (days^{-1}) or doubling/generation times (days),

which should be possible with your dataset with both 2H-PLFA and 18O-DNA data. This would help readers by allowing comparison of your data to past studies.

We added this calculation and reported estimate rates in the results section (L. 192-196 and 200-206). The equations used for calculations were added in the methods section (eq. 5 and 10).

Lines 257 – 271: Is it known whether fungal storage components (triglycerides) use the same FAs as fungal membrane biomarkers (PLFAs)? If so, how do we know that NLFAs measured in this study are not just hydrolyzed PLFAs? If not, what NLFAs were detected and how do they differ from PLFAs?

We have previously shown that the specificity of PLFA and NLFA is maintained when extracting them from organisms grown in multiple pure cultures (reference 40: Gorka et al., 2023). We added this clarification to the manuscript:

L. 137-139: *“As the NLFAs maintain the same taxonomic specificity as PLFA, they can be used in the same way to differentiate microbial groups.”*

We cannot rule out that a minor part of newly produced PLFAs are lost via cell death and hydrolysed, and then measured as NLFA. However, this is very unlikely to happen in the short time of the incubation (48 hours), not only because total bulk microbial community turnover was reported to be around only 0.3 to 7% (Pold et al., 2020), but also because PLFA contribute with two fatty acid chains (as compared to three from triglycerides). This makes us confident that the potential contribution from PLFA to NLFA is minor. To fully address this potential minor source of uncertainty we added a sentence to summarize our answer in the manuscript:

L. 383-385: *“Similarly, PLFA degradation could cause some newly formed fatty acids in PLFA to be measured within the NLFA pool after cell death and phosphate head removal. However, this contribution would be minimal given the short incubation time.”*

References:

Gorka, S. *et al.* Beyond PLFA: Concurrent extraction of neutral and glycolipid fatty acids provides new insights into soil microbial communities. *Soil Biology and Biochemistry* **187**, 109205 (2023).

Pold G. *et al.* Heavy and wet: The consequences of violating assumptions of measuring soil microbial growth efficiency using the ¹⁸O water method. *Elementa: Science of the Anthropocene* (2020)

Discussion

Lines 314 – 320: I am unsure what you mean by “PLFA composition alone (based on PLFA abundance) does not reliably describe the active fraction of the microbial community.” Do you mean that (1) the PLFA profiles do not match with what one would expect from sequencing, or (2) that greater PLFA abundance does not necessarily indicate greater growth rates? If (1), how are you determining the active fraction of the community without sequencing the DNA recovered? An approach similar to Koch et al. 2018 <https://doi.org/10.1002/ecs2.2090> would be required. If (2), a recent paper noting the decoupling of lipid-production and abundance is missing from this section (reference 35 (duplicate reference 80)) and should be discussed here as your conclusions are in agreement. In either case, a plot comparing PLFA/NLFA abundance with isotopic enrichment would be useful.

Thank you for pointing out this unclarity. We were trying to convey that indeed the abundance of PLFA markers does not clearly describe the active community at a certain time point (reviewer point n. 2).

We now clarified the sentence in L. 338-340. We also added a graph in the supplementary to show that greater abundance does not indicate greater growth rates (reported here below as well), as well as its discussion in the result section (L. 231-233).

Lines 316 – 317: “While PLFAs can have a fast turnover time in soil, our results indicate that this might not hold true during drought conditions.” I would believe that PLFAs degrade more slowly in drought conditions, but I am unsure how your data proves this. You only measure PLFA production; how do you infer that PLFA degradation is slowed during drought? I don’t think this can be established by production rates alone.

We agree with the reviewer that further data would be necessary to prove this, and we therefore removed the statement from the manuscript.

Lines 394 – 396: It seems odd to me that a water assimilation factor is applied for lipids but not DNA. Is this the case? Are there estimates of water assimilation factors for DNA? I would believe that varying the a_w value for lipids affects the growth calculations, but wouldn’t this be true for DNA as well? Applying a factor (even an estimate) could affect your conclusions regarding the similarities/differences of growth rates measured between these methods.

We agree with the reviewer that an assimilation factor should be used for DNA but to our knowledge none is currently available or used. The aim of the manuscript was not to understand the reason for the differences in absolute values between the two methods but to highlight that they capture the same relative differences induced by the climate change treatments. We added a sentence in the discussion to make the reader aware that DNA is not corrected for an assimilation factor in L. 427-428: “An assimilation constant is not currently used for DNA, making the comparison between the two methods difficult.”

Methods

Lines 542 – 545: See points above regarding attributing compounds to certain taxa. Review by Willers et al. 2015 <https://doi.org/10.1111/jam.12902> may be useful here (and useful to cite).

We thank the reviewer for the suggestion, and we added the suggested reference.

Lines 504 – 628: Total microbial biomass is measured via chloroform fumigation extraction (CFE), a standard method for determining “total microbial biomass carbon” in a soil. However, recent work (Mason-Jones et al. reference 30) suggests that CFE may be an underestimate because it overlooks storage compounds including triglycerides. So, it is unclear whether the calculations of Growth_NLFA (line 569) are accurate. Multiplying $(\text{NLFA C}_{\text{produced}})/(\text{Total NLFA C}) * \text{Microbial BMC} / \text{time}$, where Microbial BMC does not include triglycerides, seems risky, but I may be missing something. At least, this feels like a caveat that should be explicitly stated.

We did not calculate Growth_NLFA by multiplying with the results from CFE, this was only done for the PLFA (L. 635).

Throughout: As the authors say, the extent of isotopic labeling is determined relative to the isotopic enrichment of the ^{18}O and ^2H tracer solutions. They accomplish this by generating equilibration curves (Supplementary Text/ Fig. S8) and integrating the isotopic enrichment over 48 hours, taking the average value as the effective isotopic tracer strength. Similar to point #1, it is necessary to state upfront the values of the effective label strength and how they varied across the soil conditions. It is also necessary to show raw $^2\text{H}/^{18}\text{O}$ enrichment of PLFA/NLFA over time so that readers can assess where the greatest rate of enrichment occurred.

We added the average isotopic enrichment of the soil water (between 12.6 and 18.9 at% for ^2H and between 18.9 and 22.7 at% for ^{18}O) for the different assays experiment in the text in L. 541-549, for each treatment (the generated equilibration curves had already been shown in the supplementary figure section). Also, in response to an earlier comment, we added the raw data on the atom percent for PLFA, NLFA and DNA as supplementary data. We only measured PLFA/NLFA/DNA enrichment after the end of the incubation (48 hours) and did not collect several time points, therefore $^2\text{H}/^{18}\text{O}$ enrichment in fatty acid or DNA over time cannot be shown.

Throughout: I am also curious the extent to which vapor equilibration homogeneously labels the soil. I imagine water would penetrate from outside to inside of soil aggregates and therefore soil microbes existing on the periphery of an aggregate would be labeled more rapidly than those in interior spaces. Could you comment on whether this can be assessed and the degree to which it could play a role in your measurements?

It is difficult to specifically comment on this. We have been testing this method in multiple recent laboratory experiments, including a test of sieved compared to soils with intact core structure and found no differences in equilibration curves. This would imply that large aggregates do not really affect the speed of equilibration, but that this depends on other controls. While it is difficult to conclude whether there are significant differences between outside and inside aggregates, it remains a possibility. We still aimed at measuring an aggregated measure (and not spatially explicit measurement of growth), and therefore we think that this potential minor effect would not make substantial differences, especially given the aim of the manuscript to analyse differences in treatment effects on the same soil type. We agree

that this could be a very interesting follow up study that would however require a significant different experimental setup.

Line Edits

Line 40 – abstract: “Community-level CUE (the balance between anabolism and catabolism) was unaffected by drought but decreased in future climate conditions.” The wording in this line is unclear. One would expect that drought is a possible future climate condition for many biomes. I know this is further discussed, but for the abstract, it would be good to clarify what the future climate condition entails that decreased CUE. In addition, a CUE decrease may not be intuitive to readers unfamiliar with the term – does a decrease favor anabolism or catabolism? It would be helpful to specify that here.

We modified the following sentence in the abstract (L 42-43) to clarify this statement.

Line 122 – abstract: “all microorganisms synthesize lipids and fatty acids to build and maintain cell membranes regardless of their metabolic activity and cell cycle stage” is a bit misleading. While 2H-lipid-SIP quantifies anabolic activity by tracking the production of membrane lipids, this production rate does likely relate to metabolic activity and the metabolic production of free energy on some level. Perhaps it is more accurate to say, “all microorganisms synthesize lipids and fatty acids to build and maintain cell membranes regardless of their metabolic strategy and cell cycle stage.”

We agree and we modified the sentence as suggested by the reviewer (now L. 125).

Supplementary Figure S10: In my view, the more appropriate comparison here is not a condition-specific paired t-test, but rather a comparison of the 1:1 line and a linear regression of the data plotted in panel B. To my eye it looks like there could be an offset between 1:1 line and a line of best fit through the data. I am not very concerned with deuterium toxicity at the enrichments applied in this study, but this would be useful to plot in order to prove it to readers, especially because your deuterium enrichments are relatively high.

We plotted the line as suggested by the reviewer. The plotted line deviates slightly from the 1:1 and we added a description of this to the text (L. 552-554). Also, a previous study suggested that the amounts we used should not cause strong changes in activity as no effect of deuterium on microbial activity was found up until 50 at% (Warren, 2023).

Warren CR. 2023. LC-MS analysis of D2O-labelled soil suggests a large fraction of membrane lipid exists within slow growing microbes. *Soil Biology and Biochemistry* **177**: 108912.

Suggestions for reframing

The major concerns I have outlined above, in my opinion, should preclude this work from publication in its current form. However, I do not want to imply that I see no value in the experiments conducted here. The authors could benefit from stepping back from taxonomically-resolved estimates of growth (e.g., fungi vs. bacteria, gram+ from gram-, etc.) and focusing on the community level measurements. How do the community-level growth rates compare to other lipid-SIP and DNA-SIP estimates of growth (see meta-analysis by Koch et al. 2018 for example).

How do the values derived from the vapor-SIP method compare to studies where the soils are rewetted or slurried? How does the drought condition compare to previous studies of soil rewetting? These are all interesting scientific questions that can be answered with the dataset in hand and would be valuable for the community. Such a manuscript would be appropriate for a targeted soil microbio or environmental microbio journal.

The comparison of community-level production of PLFAs and DNA, inferred from 2H and 18O, also provides opportunities to discuss soil microbial physiology, specifically how growth is manifest in this system and balanced between nucleic acids and membrane lipids. More discussion here is warranted regarding how these rates relate to each other and to respiration. Indeed, it does seem that there exists substantial offset between PLFA and DNA growth rates. This is intriguing! Could this difference point to differences in biomolecular investment? Or does this speak to the nature of the methods: DNA is replicated in a more binary fashion than lipids are produced (more continuously). Focusing on these aspects would pull the scope of the paper away from the more fraught/risky aspects of taxon-specific growth rate assignment and more towards a comparison of DNA and lipids as growth markers at the community-level.

We thank the reviewer for all the suggestions, which points out that the method offers a high potential and could be applied for many different experimental studies. However, as stated above, the aim of this current manuscript was not merely to compare methodologies but to assess differences in treatment effects of simulated climate change scenarios. We believe, with the clarification provided above, our approach, results and interpretation remain valid.

Furthermore, the suggestions put forward by the reviewer have already been analysed in previous published manuscripts:

1) A recent paper in PNAS already compares results obtained by different methods that measure turnover time in soil samples (Caro et al., 2023);

2) We already discussed how the vapor method compares to the standard method in dry vs moist soil in terms of community growth and CUE results (Canarini et al., 2020).

While we appreciate the suggestion to reframe the manuscript, we believe our approach effectively addresses the study's aims. We have modified some parts in the introduction to emphasize and clarify our aims (L. 156), as well as restructured some parts of the results section (L. 187-206). The main novelty of our study is the combination of the experimental approach and the unique field experiment simulating different climate change scenarios, and we believe that our insights into community and group specific responses represent novel and unique findings.

References:

Caro, T. A., McFarlin, J., Jech, S., Fierer, N. & Kopf, S. Hydrogen stable isotope probing of lipids demonstrates slow rates of microbial growth in soil. *Proceedings of the National Academy of Sciences* 120, 9 (2023).

Canarini, A. et al. Quantifying microbial growth and carbon use efficiency in dry soil environments via ^{18}O water vapor equilibration. *Global Change Biology* 26, 5333–5341 (2020).

Reviewer #3 (Remarks to the Author):

This paper by Canarini et al. investigates the response of soil microorganisms to future climate scenarios, including increased atmospheric CO₂ concentration, increased temperature, and drought. Using a novel isotope-based method, the authors can partition microbial growth rates into bacterial and fungal growth rates and explore their different responses to environmental change. The authors also explore the carbon allocation of microbial communities into storage compounds in the different climate change scenarios.

The authors use an innovative method that allows them to answer an important question: how do fungi resist long periods of drought? Lots of literature suggests that fungi are more resistant than bacteria to drought, however not many studies show the mechanisms regulating this, which Canarini et al. do. Overall, I found that the manuscript was well written, clear, and easy to follow. The figures and legends are clear, and so is the text.

I have some main concerns/questions about the study:

1) Authors express all their growth and respiration rates as mass specific rates, and I am unsure why is that. I believe that if the contribution of microbes to C fluxes wants to be understood the rate should be an absolute rate. Treatments might affect biomass and functions differently and thus it is difficult to disentangle effects and understand how the environment is driving microbial mediated C cycling. So, the changes in mass specific rates are they due to changes in rates or changes in biomass? Or both? I suggest at least indicating mass specific rates as well as rates per g of dry soil, followed by a discussion of the differences observed.

We agree with the reviewer that this is an important point. The rationale to use mass-specific rates is that we are primarily interested in the microbial (physiological) response to climate change, not in ecosystem fluxes, that we did not measure. Such physiological responses would be masked by changes in microbial biomass between treatments.

To address the reviewer's comment, we added data on changes in growth rates (per g of dry soil) in the supplementary section (now Fig. S8) and in the text (L. 265-266). Results expressed in this way show almost identical treatment effects when compared to mass-specific growth rates.

2)The authors incubate samples at field temperature. How can we know that the results they see is not an effect of the differences in temperature? Have the authors account for that effect?

We aimed to keep the temperature differences constant between the ambient (no warming) and future climate conditions (warming+eCO₂) to assess the growth pattern of microorganisms under the respective climate treatment. While incubating the soils at the same temperature would have allowed us to see whether different communities of microbes behave differently at the same conditions (i.e., differences in genetic adaptation to climate change or changed community composition), this was not the aim of our study. To address the reviewer comment and clarify this point to the readers we added a discussion in the text:

L.405-407, *“Furthermore, in our experimental approach we used field temperature differences to represent conditions as close to the field as possible, and therefore measured growth rates were also affected by these conditions and not only by previous exposure to the climate change treatment.”*

Furthermore, we warn the readers that while measured effects of future climate treatment could be very important, they also depend on the season, as found in a previous study (Simon et al., 2020), which should be kept in mind when interpreting the results of our future climate treatment (L. 404).

3) I would like to see the continuous data of the field experiment rather than the end point moisture level (Table S6), to understand the moisture history the soil microorganisms have been exposed to.

We have added information on soil water content development over time in the results section (L184-187) and as Fig. S1 in the supplementary section (Fig S1, see graph also below).

In line comments:

Line 83. CUE is defined as C allocated to growth relative to respiratory processes. Perhaps, C allocated to growth relative to the total amount of C used by microorganisms would be more appropriate.

This is correct, we have changed this accordingly.

Line 85. I understand why authors here refer to growth. However, there is not a clear link back to CUE. If the aim is to reach a broader audience that might not be familiar with the this might be a bit confusing for the reader.

We have modified the sentence to clarify the link between growth and CUE:

L: 86-87: *“Improving our ability to accurately quantify soil microbial growth will help to better predict microbial CUE.”*

Lines 93-94. What happens with growth when temperatures increase?

We have modified the sentence now in:

L. 94-95 *“Warming can increase plant productivity, but also directly stimulate microbial physiological activity (growth and respiration), decrease microbial CUE and accelerate soil C losses.”*

Line 93. In which direction? Please provide examples.

We are unsure what the comment specifically refers to. The sentence provides directions as well as examples. The sentence has now been modified in response to the previous comment (see above).

Lines 101-102. Soil functions and microbial communities. Which functions and aspects of the microbial communities are the authors referring to? If the aim here is to point out a research gap I advise to be more specific.

We thank the reviewer for the suggestion, and we changed “soil functions” to “soil biogeochemical cycling” to be more specific (L. 103).

Figure 1 and in general. Why are the medians indicated instead of the average?

Boxplots are generally used together with medians when the interquartile ranges are shown, in order to give a better representation of the data distribution (the mean can be distorted more easily by outliers). We consider this way of showing the data, altogether with the reported statistic a better way for the reader to fully evaluate the results from the graphs.

Line 281-282. Please add reference.

We added the following references (now L. 306):

Bradford MA, Wieder WR, Bonan GB, Fierer N, Raymond PA, Crowther TW. 2016. Managing uncertainty in soil carbon feedbacks to climate change. *Nature Climate Change* 6: 751–758.

Kallenbach CM, Frey SD, Grandy AS. 2016. Direct evidence for microbial-derived soil organic matter formation and its ecophysiological controls. *Nature Communications* 7: 13630.

Wang C, Qu L, Yang L, Liu D, Morrissey E, Miao R, Liu Z, Wang Q, Fang Y, Bai E. 2021. Large-scale importance of microbial carbon use efficiency and necromass to soil organic carbon. *Global Change Biology* 27: 2039–2048.

Line 300-301. How severe/intense is the drought for the microbial communities in this ecosystem?

We added the gravimetric water content to the result section (L.184-187) as well as shown the soil moisture variation (vol%) during the experimental period in a new supplementary figure (Fig. S1). Moreover, in the discussion we emphasized again the relative decrease in soil water content in the drought-treated compared to the respective controls in order to contextualize the extent of the drought treatment (now L. 318).

Line 364-365. This could be due to the time frame where the recovery was measured. In addition, did authors measure growth and respiration immediately after sampling? That could also have an effect. Please clarify how much time passed between sampling and measuring.

We agree with the reviewer that the time frame is very important following rewetting conditions and results might have been different if measured right after rewetting. The time between the rewetting and the start of the incubation was three days and we used 48 hours of incubation. Therefore growth rates were measured effectively five days after rewetting. We now clarify this (L. 394-395) and added the sentence:

“measured five days after rewetting to avoid immediate respiration responses to rewetting right after water additions”.

Line 404. “During” has been written twice.

Thanks for spotting this, we corrected the text.

Lines 445-446. Please clarify how long between sampling date and measuring dates (see above).

Incubations were started 24 h after sampling. The information was given in L. 506.

Lines 475-476. Why did authors decide to incubate at different temperatures instead of a single common temperature? How can authors disentangle the direct temperature effect from the field treatment effect? (See above)

Our aim was not to disentangle direct temperature effect from treatment effect but rather to target growth rates as close as possible to real conditions, and therefore we decided to keep the temperature difference as measured in the field.

Lines 510-512. This is not clear, which specific growth rates have to be considered? Please clarify.

We apologised for the confusion, we modified the word and the sentence now reads:

L 563: “Fatty acid biosynthesis and subsequently also ²H incorporation combine fatty acid production related to membrane growth, but also membrane repair; therefore, calculated growth rates need to be considered accordingly.”

Line 545. It is not clear whether authors use actinobacteria as Gram + bacteria when looking at fungal to bacterial ratios.

We rephrased the sentence to clarify that Gram+ also contained actinobacteria and were used to calculate fungal to bacteria ratios (now L. 608-612).

REVIEWER COMMENTS

We thank the two reviewers and the editor for providing a chance to further improve and strengthen our manuscript.

Our point by point answers are given in blue below each comment.

Reviewer #1 (Remarks to the Author):

The authors have done a great job revising their manuscript. The manuscript will make a nice contribution to the literature. Congratulations!

We thank the reviewer for the nice comment and for the invaluable suggestions in the previous round of review, which have helped to improve the manuscript.

Reviewer #2 (Remarks to the Author):

Canarini, Fuchslueger, et al.,

The authors have made substantial progress on this manuscript. The authors have elegantly addressed the majority of my concerns, and I commend them for creating an eloquent and important piece of literature.

Regarding my suggested reframing of the manuscript, the authors have sufficiently defended their current framing and provided much needed methodological context and caveats to their main text. I can accept the current framing of the revised paper and thank the authors for their diligence in addressing my comments.

I believe the majority of my comments have been addressed, including all of my minor and line comments. However, I must continue to raise concerns regarding my major comments No. 1 and No. 5. I do not think my remaining concerns preclude this work from publication, but I would like to see them addressed, if possible, before this work enters print.

We thank the reviewer for the positive evaluation of our revised manuscript, and also for the invaluable suggestions that have strongly improved the manuscript. We implemented the requests of the reviewer and addressed them below.

Major Comments:

1. Inclusion of labeling data.

“We now include further information both on natural abundance and $2\text{H}/18\text{O}$ tracer incorporation in the supplementary materials ...”

Thank you for the inclusion of the average isotopic composition data. This will be useful for readers in contextualizing the extent of labeling. While I think this data is useful to include in the main text, it doesn't address the point at-large. Readers and those designing similar experiments, I think, need to know how specific compounds (individual lipids measured, DNA) became enriched over time (even if that is two timepoints, start and end). An average value, with no corresponding figure, does not convey heterogeneity in compound-specific labeling and may be obscuring highly labeled compounds or poorly labeled compounds. I see the authors

have included their compound-specific labeling data as a supplementary dataset, but I would suggest a figure or SI figure, so that this data is more central to the story.

We added a figure showing compound specific label distribution in the supplementary section (now Fig. S 14). We report the figure here below for clarity.

Figure S14: Compound-specific isotope composition of the FAMES included in our study given in ^2H atom% shown during a) 'Drought' and b) the 'Recovery' period. The data is grouped by treatment (Ambient, Drought, Future Climate and Future Climate + Drought). Values are shown as average ^2H atom % (error bars show standard errors, the sample size represents biologically independent samples; $n=4$).

2. Free fatty acids in the environment.

"The applied derivatization method does not derivatize free fatty acids and therefore those are excluded from the analysis"

Thank you for clarifying this in the main text and providing citation. The authors have addressed this concern.

We thank the reviewer to give us the chance to clarify this.

3. Slower triglyceride degradation.

"We agree with the reviewer that we would underestimate mass-specific NLFA production rates due to possible necromass-NLFA. However, in Fig.4b we present results of total NLFA produced (total amount) as percentage of PLFA (also total amount)..."

The authors have addressed this concern by explicitly stating this caveat in the text. I would also suggest including the caveat in the figure caption (Fig. 4) as an explanation to why the NLFA/PLFA ratio is plotted in panel B. I suppose the community needs to start measuring the degradation rates of different lipid classes in soil!

We agree with the reviewer, and hope that our manuscript acts as a primer for the scientific community and stimulates further research.

We modified the figure legend in Figure 4 accordingly:

“Ratio of fungal specific newly produced NLFA to newly produced PLFA (expressed as percentage), indicating that fungi increase the relative investment in NLFAs during drought. This ratio allows to account for a potential underestimation of mass-specific NLFA production rates caused by potential necromass-NLFA accumulation.”

4. Compound identification via GC

Thank you for addressing my concern regarding the standards mixtures. I hope the authors can appreciate the alarm of seeing identified compounds that were not present in the standards. The authors have included sufficient information, including that they used GC-TOF-MS to further identify compounds. Seeing the example spectra has helped the authors prove their point, assuming this spectrum is a soil sample and not a library of mixed standards (by eye it looks like a typical soil profile...).

By addressing my concerns on compound ID via GC, the authors have also assuaged my concern of taxonomic ID. In brief, if the GC peak identification was spotty, then I would have serious concerns about their taxonomic assignment. However, because it looks like the authors have done their due diligence in building a FAME compound library and conducting MS identification, I can believe their taxonomic assignment follows the best knowledge available in the literature.

We thank you the referee to give us the chance to clarify this point.

5. Quantification of lipid abundance.

I remain concerned about the authors calculation of PLFA growth. I understand how the authors are calculating relative PLFA production without extraction and derivatization standards, as well as how the Lipid_produced and Lipid_total terms cancel out in their calculations and are multiplied by the total microbial biomass carbon (MBC) as estimated by chloroform fumigation extraction (CFE). However, the assumption the authors are making here is that lipid and biomass C (measured by CFE) is a 1-to-1 relationship. Because the authors did not include extraction standards, they cannot know with confidence how much PLFA was present in their samples per gram of soil. They only know the total MBC from the CFE. Of course, there is an intuitive link between lipid content and total MBC, but these do not exist at a one-to-one relationship and there may be substantial offset between the two (e.g., see Bailey et al. 2002 [https://doi.org/10.1016/S0038-0717\(02\)00070-6](https://doi.org/10.1016/S0038-0717(02)00070-6)). Therefore, production rates of PLFA may not adequately translate to MBC. To avoid confusion, I would advise the authors stick to relative growth rates, not normalized to mass, (e.g., day⁻¹, yr⁻¹, etc.) as their primary reported metric. The authors have stated that their intention is to focus on relative differences between the treatments, which I think is more readily accomplished with intuitive units of measurement that are already widely reported in the literature.

We have included data expressed in day⁻¹ in the main part of the manuscript, now Fig. 1 (reported here below for clarity) and moved the mass specific growth rates converted to microbial carbon based on CFE data in the supplementary section (Fig. S2). We have changed the figure legends accordingly (see lines 211-217) and we have added the relevant calculations

in the materials and methods section and the relevant figure citation in the result section (see lines 189, 650, 701).

Minor comments:

Please check the rendering of your equations, some exhibit blank spaces.

We thank the reviewer for pointing this out. We have double checked the pdf version and all equations should be ok now.